# Two RNA binding proteins, ADAD2 and RNF17, interact to form a heterogeneous population of novel meiotic germ cell granules with developmentally dependent organelle association

**Lauren G. Chukrallah, Sarah Potgieter, Lisa Chueh, Elizabeth M. Snyder** *

Department of Animal Science, Rutgers, The State University of New Jersey, New Brunswick, New Jersey, United States of America

* elizabeth.snyder@rutgers.edu

## Abstract

Mammalian male germ cell differentiation relies on complex RNA biogenesis events, many of which occur in non-membrane bound organelles termed RNA germ cell granules that are rich in RNA binding proteins (RBPs). Though known to be required for male germ cell differentiation, we understand little of the relationships between the numerous granule subtypes. ADAD2, a testis specific RBP, is required for normal male fertility and forms a poorly characterized granule in meiotic germ cells. This work aimed to understand the role of ADAD2 granules in male germ cell differentiation by clearly defining their molecular composition and relationship to other granules. Biochemical analyses identified RNF17, a testis specific RBP that forms meiotic male germ cell granules, as an ADAD2-interacting protein. Phenotypic analysis of *Adad2* and *Rnf17* mutants identified a rare post-meiotic chromatin defect, suggesting shared biological roles. ADAD2 and RNF17 were found to be dependent on one another for granularization and together form a previously unstudied set of germ cell granules. Based on co-localization studies with well-characterized granule RBPs and organelle-specific markers, a subset of the ADAD2-RNF17 granules are found to be associated with the intermitochondrial cement and piRNA biogenesis. In contrast, a second, morphologically distinct population of ADAD2-RNF17 granules co-localized with the translation regulators NANOS1 and PUM1, along with the molecular chaperone PDI. These large granules form a unique funnel-shaped structure that displays distinct protein subdomains and is tightly associated with the endoplasmic reticulum. Developmental studies suggest the different granule populations represent different phases of a granule maturation process. Lastly, a double *Adad2-Rnf17* mutant model suggests the interaction between ADAD2 and RNF17, as opposed to loss of either, is the likely driver of the *Adad2* and *Rnf17* mutant phenotypes. These findings shed light on the relationship between germ cell granule pools and define new genetic approaches to their study.

**Data Availability Statement:** All data are in the manuscript and/or supporting information files.

**Funding:** The work described herein was supported by financial support from the Eunice Kennedy Shriver National Institute of Child Health and Human Development (NIH-NICHD F32 HD072628, K99/R00 HD083521, and R01 HD107066 to EMS) and Rutgers University (to EMS). The funders had no role in study design, data collection and analysis, decision to publish, or preparation of the manuscript.

**Competing interests:** The authors have no competing interests to declare.

## Author summary

To differentiate successfully, male germ cells tightly regulate their RNA pools. To do so, they rely on RNA binding proteins, which often localize to cytoplasmic granules. The majority of studies have focused on a single granule type, the intermitochondrial cement, which regulates piRNA biogenesis. As a result, there is limited knowledge about the other granules. Here, we identify an interaction between two RNA binding proteins, ADAD2 and RNF17, and demonstrate mutants share a rare germ cell phenotype. Further, ADAD2 and RNF17 colocalize to the same germ cell granules, which are observed as two morphologically unique types. The first subset of ADAD2-RNF17 granules associate with known granules and likely play a role in the piRNA pathway. The second granule type forms a unique shape with distinct protein subdomains and may play a role in regulating translation. This second population is in close proximity to the endoplasmic reticulum and appears to form by relocalization and molecular remodeling of the first. Genetic models further suggest the interaction between ADAD2 and RNF17 likely drive the mutant phenotypes. These findings identify a novel population of germ cell granules that undergo maturation across germ cell differentiation and appear to regulate distinct aspects of RNA biology.

## Introduction

The male germ cell relies on complex RNA biology for successful differentiation. As a result, they express and require a wide range of RNA binding proteins (RBPs), many of which are housed in non-membrane bound, cytoplasmic organelles termed germ cell RNA granules or germ cell granules. These granules are especially prevalent during meiotic and post-meiotic germ cell differentiation and are fundamental for proper developmental progression in multiple species [1,2]. In mammalian male germ cells for example, six types of granules have been identified via electron microscopy (EM), five of which can be found in meiotic spermatocytes [3]. Loss of core granule proteins commonly leads to meiotic or post-meiotic germ cell arrest [4–8], underscoring their importance in germ cell differentiation and male fertility.

Historically, the function of these granules has been defined primarily by functional knowledge of associated proteins or subcellular localization. One particularly successful example of this is the intermitochondrial cement (IMC), which is an amorphous matrix distributed between the mitochondria of meiotic male germ cells (spermatocytes) with well-defined protein and RNA composition [6,9–11]. Based on a combination of genetic and functional analyses, the IMC has been identified as a primary site of biogenesis for piRNAs (PIWI-interacting RNAs) [7,9,12–14], an abundant class of small non-coding RNAs that modulate mRNA translation and stability [15,16] and are required for transposon silencing [17,18]. Many IMC-localized proteins, including the primary piRNA binding proteins PIWIL1 [19] and PIWIL2 [17] have distinct impacts on the RNA biogenesis events localized to the IMC [7,14].

Unlike the IMC, the other meiotic germ cell granules have less defined compositions or functions. Similarly, whether they have distinct associations with any intracellular organelles has not been thoroughly explored. Three (the satellite or sponge body—SB, loose aggregate strands–LAS, and irregularly shaped perinuclear granules- ISPGs) are known to contain RBPs associated with mRNA storage and translation regulation such as DDX4, DDX25, and NANOS1 [20–22]. Further, ISPGs have been associated with the smooth endoplasmic reticulum while both the SB and LAS are described as cytoplasmic. In even greater contrast to the

IMC is the one entirely unstudied meiotic germ cell granule, known only as the "cluster of 30-nm particles", which has no defined resident proteins and has only been observed via electron microscopy. In addition to relatively poor composition and functional knowledge for the non-IMC granules, the exact relationship between them has never been defined [23]. Similarly, whether the cluster of 30-nm particles or any of the other EM-defined granule types represent single, molecularly homogenous populations has been almost entirely unexplored. Together these questions represent a long-standing mystery in male germ cell RNA biology.

One promising approach to address these mysteries involves detailed studies of RBPs that form spermatocyte germ cell granules. Of particular interest are those proteins that have yet to be assigned to a specific granule population. ADAD2 (adenosine deaminase domain containing 2), a testis specific RBP [24], forms a spermatocyte germ cell granule and is required for successful male germ cell development as *Adad2* mutant males are completely infertile, with germ cell development halting abruptly during post-meiotic germ cell differentiation [24]. Appearing first in pachytene spermatocytes wherein it is largely cytoplasmic, ADAD2 coalesces into a distinct perinuclear granule during mid-meiosis and remains thus through the end of meiosis. Little is known about the composition of the ADAD2 granule beyond its lack of DDX25 [24], known to mark all the spermatocyte germ cell granules excluding the cluster of 30-nm particles [21,25]. Functionally, ADAD2 has been implicated in transcript-specific translation regulation [26]. However, whether ADAD2's granular localization is related to its role as a translation regulator is unclear.

To define the ADAD2 granule, we set out to identify additional protein components and relate the granule's composition to other meiotic granules. Using the mouse as a model and leveraging multiple single and complex genetic models as well as high-resolution imaging modalities, we further dissect the timing and nature of granule formation as well as the role of the individual proteins. Together, these studies describe a population of novel meiotic germ cell granules that may play a unique role in the complex RNA biology of the germ cell.

## Results

### ADAD2 interacts with RNF17, a testis-specific RNA binding protein

The ADAD2 granule appears to be distinct from the best characterized spermatocyte granule, intermitochondrial cement (IMC) [24]. To better determine the molecular nature of the ADAD2 granule, we immunoprecipitated ADAD2 from wildtype (n = 3) and *Adad2* mutant (*Adad2* $^{M/M}$, n = 1) testes at 42 days post-partum (dpp) followed by mass spectrometry (IP-MS) to identify potential ADAD2-granule associated proteins (Fig 1A and S1 Table). Hits identified in all wildtype samples but not the mutant included well characterized post-meiotic germ cell proteins along with several RNA binding proteins (Fig 1B). Of the significant peptides identified in wildtype only, ADAD2-derived peptides represented nearly a tenth, confirming efficacy of pulldown. However, the highest number of peptides identified belong to another RNA binding protein, ring finger protein 17 (RNF17). RNF17 peptides comprised over a fifth of those identified as significant. RNF17, like ADAD2, is testis-specific and has been reported to form a spermatocyte granule [8].

To confirm the ADAD2-RNF17 interaction, immunoprecipitation of either ADAD2 or RNF17 in an additional set of 42 dpp wildtype testes as well as in *Adad2* mutant [24] and *Rnf17* mutant (*Rnf17* $^{M/M}$) [8] testes was performed. The resulting immunoprecipitates (IPs) were probed for ADAD2 and RNF17 (Fig 1C). As expected, IP of either ADAD2 or RNF17 in wildtype testes resulted in robust detection of the precipitated protein. In the case of RNF17, this includes a large and small protein isoform (RNF17L and RNF17S), both of which have been detected previously [8]. Further, mutation of either *Adad2* or *Rnf17* resulted in a

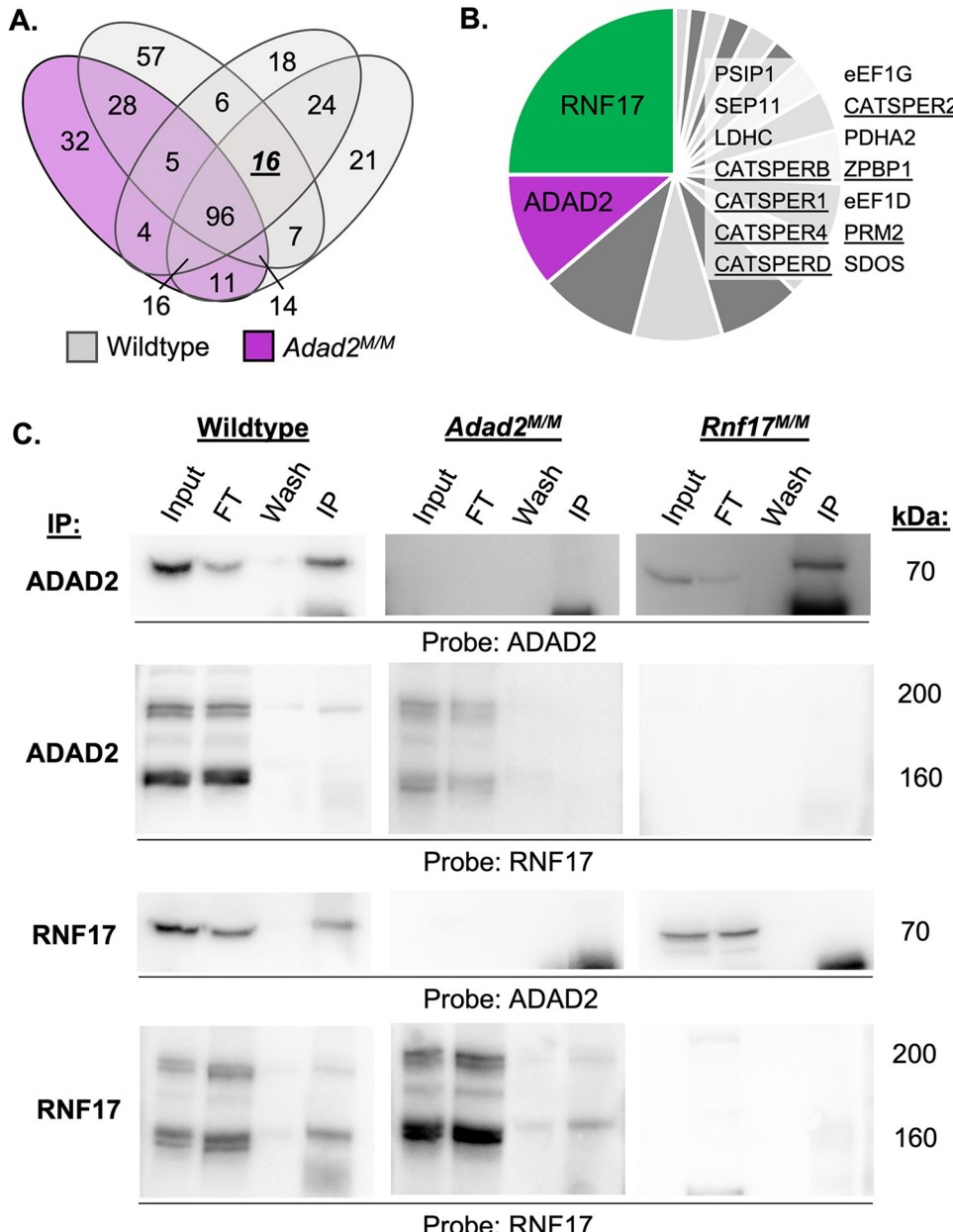

**Fig 1. ADAD2 interacts with another RNA binding protein, RNF17. A.** Number of mass spectrometry-identified proteins from ADAD2 immunoprecipitation (IP) of whole testis lysate from 42 dpp wildtype (grey ovals, n = 3) or *Adad2^{M/M}* (purple oval, n = 1). **B.** Summary of proteins detected across all wildtype IPs but not *Adad2^{M/M}*. RNF17, protein of interest, in green, and ADAD2, for confirmation of genotype and IP, in purple. Table inset includes remaining 14 proteins identified in all three mutants but not wildtype. Underline indicates well-known post-meiotic germ cell proteins. **C.** Confirmation of ADAD2-RNF17 interaction via immunoprecipitation-Western blotting in wildtype, *Adad2^{M/M}*, and *Rnf17^{M/M}* 42 dpp whole testis lysates demonstrating a specific interaction between ADAD2 and RNF17L. 'Input'—total testis lysate, 'FT'–flow-through (unbound lysate proteins), 'wash'—first wash of beads post binding, 'IP'—immunoprecipitate. Blots representative of results from at least three IPs per genotype. Approximate molecular weight reported for each band.

complete loss of ADAD2 or RNF17 detection, respectively. As expected from the IP-MS analysis, IP of ADAD2 resulted in definitive isolation of RNF17, specifically the large isoform RNF17L, while IP of RNF17 pulled down ADAD2. Lastly, mutation of either *Adad2* or *Rnf17*

abrogated IP of the other. Together, these targeted IP studies confirm the initial IP-MS results and demonstrate ADAD2 and RNF17L interact *in vitro*.

To assess the impact of either *Adad2* or *Rnf17* mutation on the abundance of the other, we next quantified ADAD2 and RNF17 abundance in 42 dpp wildtype and mutant testes (S1A Fig). This analysis revealed loss of ADAD2 led to a reduction but not loss of RNF17, confirming the IP-MS results were not a function of protein loss in the mutant. Similarly, RNF17 loss led to reduction but not loss of ADAD2 protein along with the appearance of a smaller ADAD2 protein isoform. Given mutation of both *Adad2* and *Rnf17* results in altered cellularity at 42 dpp, we performed a similar analysis at 21 dpp (S1B Fig), a time point at which neither model should have substantial changes in cellularity. In contrast to the observations in 42 dpp testes, loss of ADAD2 lead to a slight increase of both RNF17 isoforms while RNF17 loss had minimal impact on ADAD2 abundance. While these findings suggest ADAD2 may directly influence RNF17 by modulating protein abundance, potential ADAD2-induced changes would not impact the above observed IP-based interactions, thus confirming ADAD2 and RNF17L interact *in vitro*.

## Both *Rnf17* and *Adad2* mutants exhibit round spermatids with abnormal chromocenters

As ADAD2 and RNF17 interact, we next wondered whether *Rnf17* mutation mimics that of *Adad2*. *Rnf17* mutant males have been shown to exhibit severe post-meiotic germ cell loss culminating in total male infertility [8] and previously published analyses of *Adad2* mutant testis histology suggest a similar profile of germ cell loss during round spermatid development [24,26]. To determine whether post-meiotic phenotypes in *Rnf17* mutants mimicked that observed with ADAD2 loss, quantification of round spermatid numbers as a function of stage was performed on both models (Fig 2A). This analysis demonstrated distinct round spermatid reduction in *Rnf17* mutants similar to that observed in *Adad2* mutants. Normal round spermatids contain two distinct regions of heterochromatin, the first composed of autosomal heterochromatin and referred to as the chromocenter and the second composed of sex chromosome heterochromatin, referred to as post-meiotic sex chromatin (PMSC) [27]. The above analyses also revealed *Rnf17*$^{M/M}$ round spermatids exhibit abnormal heterochromatin as marked by regions of intense DAPI staining and the heterochromatin mark H3K9me3 [28] (Fig 2B). The heterochromatin defect observed in *Rnf17* mutant round spermatids is also observed in *Adad2* mutants. To determine if the chromatin ultrastructure defect in *Adad2* and *Rnf17* mutant round spermatids shared a similar profile, we quantified H3K9me3 foci in wildtype, *Rnf17*$^{M/M}$, and *Adad2*$^{M/M}$ round spermatids (Fig 2C). As expected, wildtype round spermatids rarely contained more than a single focus, representing a normal chromocenter associated with PMSC. However, both *Adad2* and *Rnf17* round spermatids had increased numbers of H3K9me3 foci compared to wildtype and the increase in both mutant models was similar. This effect was independent of spermatid developmental stage thus impacting the entire post-meiotic germ cell population. To date, only four [29–32] other genetic models have been reported to have similar chromocenter defects, making it unusually rare. The observation of such a rare phenotype in both *Adad2* and *Rnf17* mutants indicates they may influence similar downstream events and further suggests their interaction is biologically relevant.

## RNF17 has a distinct localization in spermatocytes that is dependent on ADAD2

Two RNF17 protein isoforms, large and small, have previously been reported [8]. However, IP of ADAD2 only detected RNF17L despite both isoforms being present in *Adad2*$^{M/M}$ samples

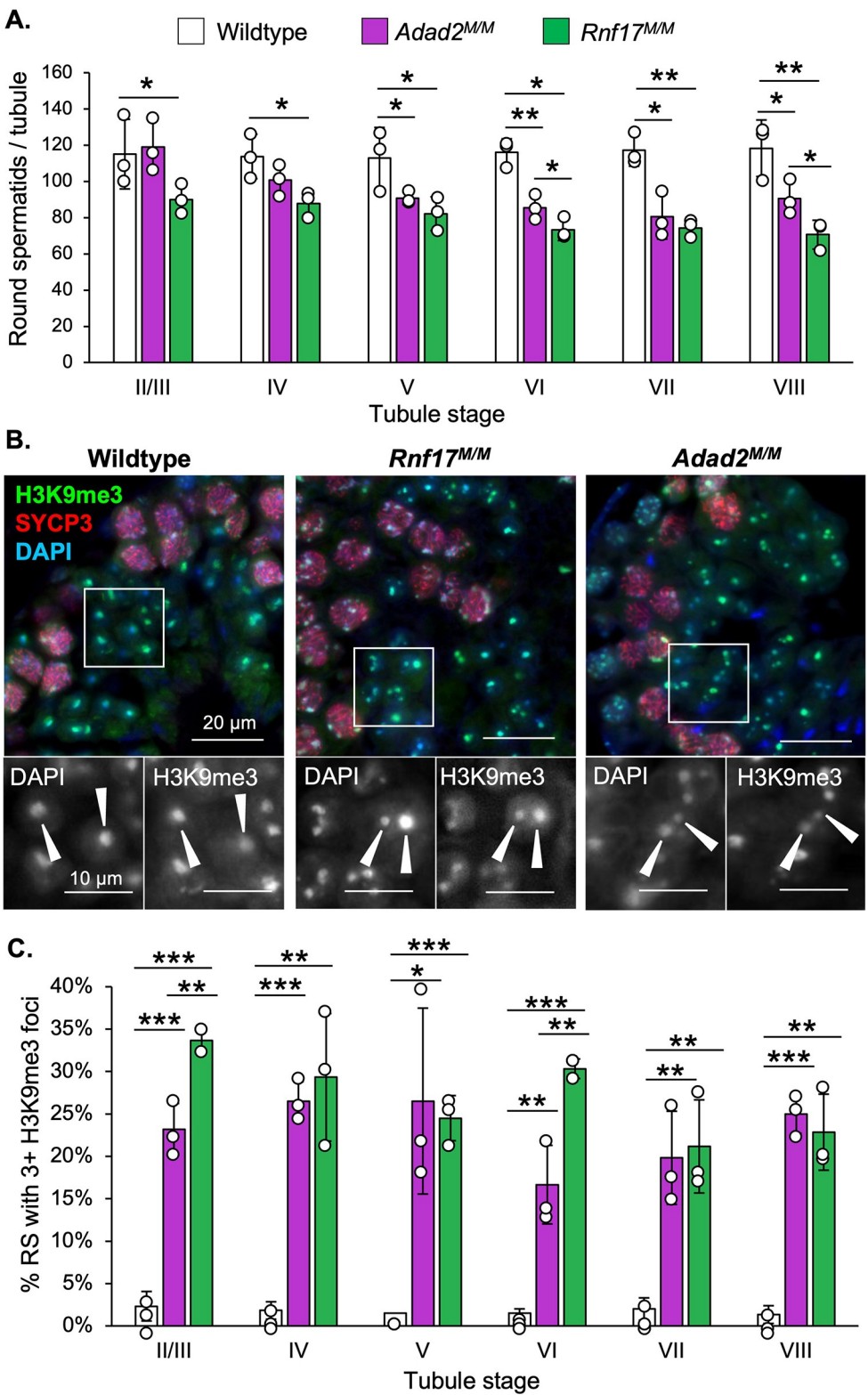

**Fig 2. Loss of RNF17 results in a distinct chromocenter phenotype also observed in *Adad2* mutants. A.** Round spermatids per tubule as a function of developmental stage in adult wildtype, *Adad2*$^{M/M}$, and *Rnf17*$^{M/M}$ testes (n = 3). **B.** Stage-matched H3K9me3 immunofluorescence in adult wildtype, *Adad2*$^{M/M}$, and *Rnf17*$^{M/M}$ testes counterstained with the stage-dependent marker SYCP3. Green–H3K9me3, red–SYCP3, blue–DAPI. 400x magnification. Insets–

DAPI or H3K9me3 signal only. **C.** Quantification of round spermatids (RS) with 3 or more chromocenter-like structures as marked by H3K9me3 staining per developmental stage in adult wildtype, $Adad2^{M/M}$, and $Rnf17^{M/M}$ samples (n = 3). Data are mean ± s.d. Significance calculated using an unpaired, one-tailed Student's t-test (*$P < 0.05$, **$P < 0.005$, ***$P < 0.0005$).

(see Figs 1C and S1A and S1B). To determine if this was a function of RNF17 availability, we identified the developmental time points both proteins are detectible (S2A Fig). This analysis demonstrated ADAD2 first appeared at 10 dpp, coincident with the first appearance of early to mid-stage pachytene spermatocytes in the developing testis. Following this, the abundance of ADAD2 increased dramatically at 15 dpp when the testis cellular profile is highly enriched for mid- to late-pachytene spermatocytes. A very similar pattern was also observed for RNF17L. In contrast, RNF17S is observed as early as 8 dpp, reaching and sustaining a maximum by 10 dpp. Together, this demonstrates ADAD2 shares a very similar developmental profile specifically with RNF17L and further suggests ADAD2's interaction specifically with RNF17L is not dependent on availability as both protein isoforms are present from 10 dpp onward.

ADAD2 forms a developmentally-regulated granule in pachytene spermatocytes [24]. RNF17L has also been described as forming a germ cell granule in pachytene spermatocytes [8]. To better define the timing of RNF17 granule formation, we examined RNF17 localization via immunofluorescence in wildtype adult spermatocytes throughout their differentiation (Fig 3A). RNF17 weakly appears in the cytoplasm of mid-stage spermatocytes (stage V) and first coalesces into small granules in stage VII spermatocytes. Following this, large cytoplasmic RNF17 granules appear between stages VIII and IX in mid- to late pachytene spermatocytes. The majority of RNF17 signal is retained in these large cytoplasmic granules until the end of meiosis (stage XII). This is similar to previous reports [8]. Comparison with ADAD2 granule formation (S3A Fig) demonstrated ADAD2 granule formation in spermatocytes is notably delayed compared to the small RNF17 granules but aligns very well with formation of the large RNF17 granule, starting in late pachytene spermatocytes of stage VIII. Similar to RNF17, ADAD2 granules are also observed in two types, one small and frequent and the other large occurring only once or twice per cell. As an alternative approach to compare ADAD2 and RNF17 granule formation, we examined them via immunofluorescence across neonatal testis development (S2B Fig). Over the course of development, both ADAD2 granule types are first observed at 15 dpp, with the appearance of mid- to late-pachytene spermatocytes. Both granule types of RNF17 were observed to appear similarly. Together, these analyses demonstrate both ADAD2 granules and the large, but not small, RNF17 granules are specific to mid- to late pachytene spermatocytes. Further, large RNF17 granules likely form just after formation of the similarly sized ADAD2 granules.

Given that RNF17 is still present in the absence of ADAD2 (S1A and S1B Fig), we sought to determine whether loss of ADAD2 impacted RNF17's spermatocyte localization. Immunofluorescence of RNF17 in $Adad2^{M/M}$ and $Rnf17^{M/M}$ mutant testes (Fig 3B) revealed that although cytoplasmic RNF17 along with small RNF17 granules can be detected in $Adad2^{M/M}$ spermatocytes, RNF17 fails to coalesce into large granules, irrespective of stage. Given the overlapping timing for the ADAD2 granule and the RNF17 granule, we next examined ADAD2 granularization in the context of RNF17 loss (S3B Fig). Like RNF17, ADAD2 fails to form large granules with RNF17 loss. In addition, ADAD2 fails to form small granules in the absence of RNF17. Together, these findings suggest the ADAD2-RNF17L interaction is required for the formation of all ADAD2 granules as well as the large RNF17 granules, both of which are specific to mid- to late pachytene spermatocytes.

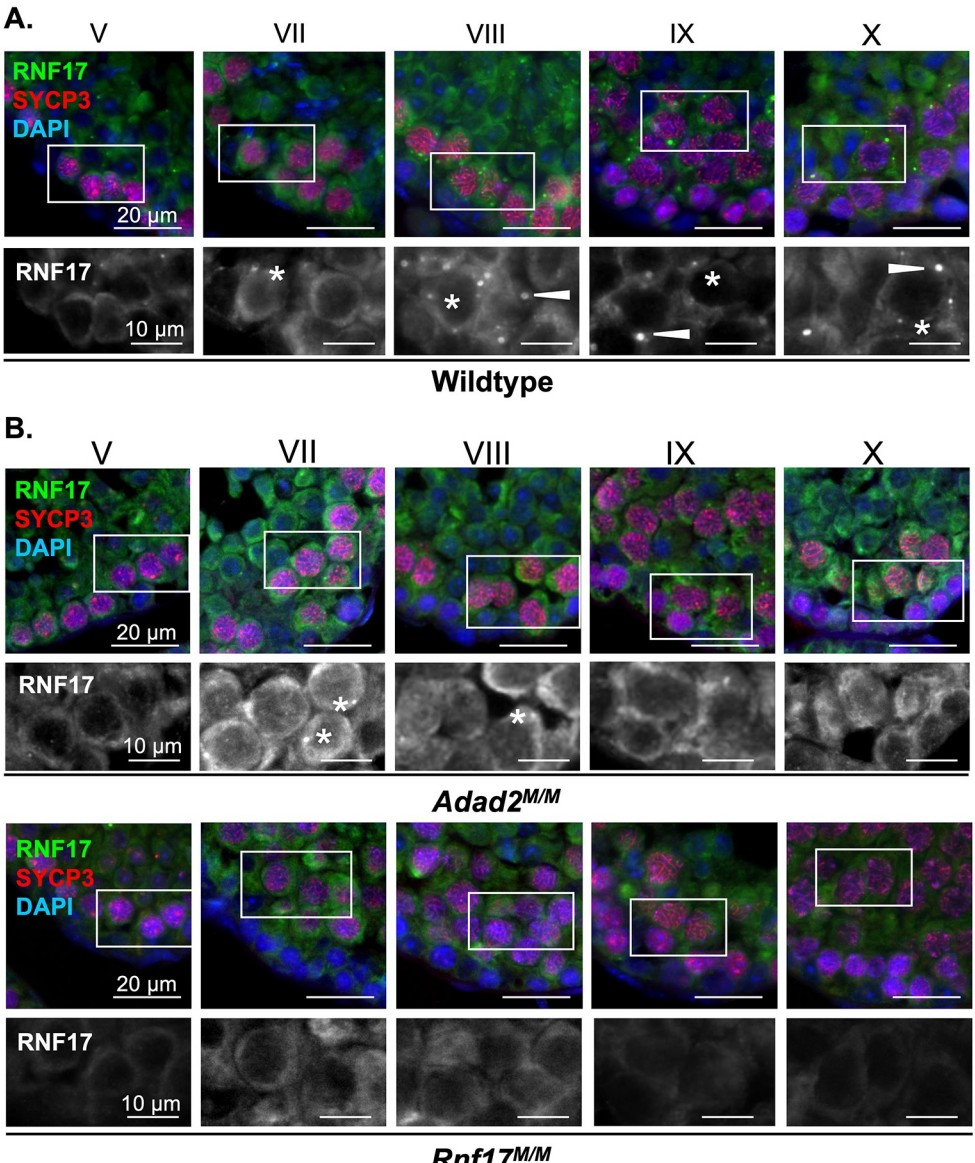

**Fig 3. RNF17 forms distinct granules in pachytene spermatocytes and requires ADAD2 for its localization.** RNF17 localization across spermatocyte development in **A.** adult wildtype testes demonstrating two phases of granule formation and two different granule types (asterisks—small granules and arrowheads–large granules) and **B.** *Adad2*<sup>M/M</sup> and *Rnf17*<sup>M/M</sup> mutant testes demonstrating RNF17's reliance on ADAD2 for formation of large RNF17 granules. Roman numerals–testis tubule cross-section stage (V containing early stage pachytene spermatocytes, VII and VIII containing mid-stage pachytene spermatocytes, IX through X containing late-stage pachytene spermatocytes). Asterisks —small RNF17 granules. Green–RNF17, red–SYCP3, blue–DAPI. 400x magnification.

## ADAD2 and RNF17 form a unique germ cell granule

ADAD2 and RNF17 share distinct phenotypic and developmental similarities. This, combined with their biochemical interaction and their reliance on one another for their granular localization, led us to wonder whether the large ADAD2 granule and the large RNF17 granule are one in the same. Immunofluorescence using fluorophore labeled anti-ADAD2 and anti-RNF17 in wildtype testes revealed near perfect colocalization between ADAD2 and RNF17 in mid- to late pachytene spermatocytes (Fig 4A). Confirmation of labeled antibody specificity was

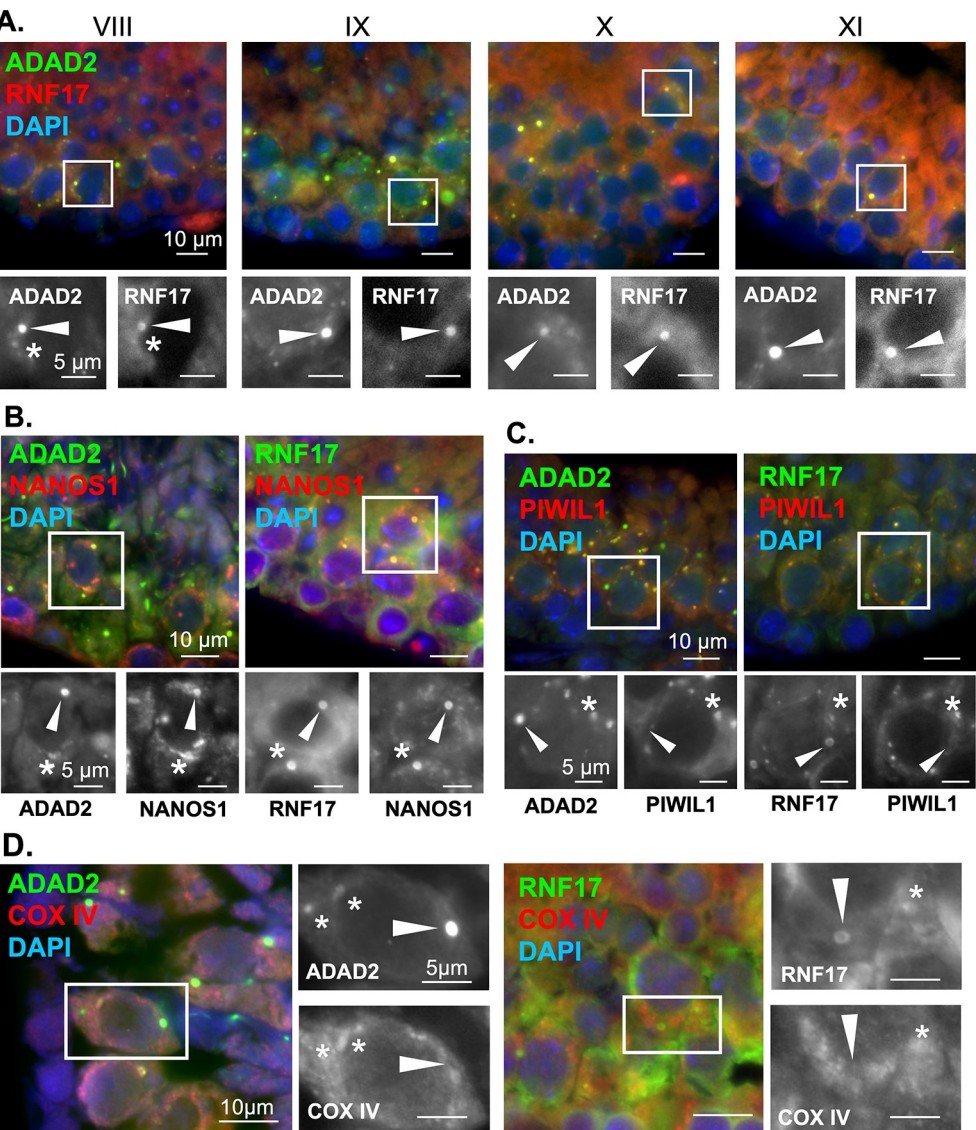

**Fig 4. ADAD2 and RNF17 colocalize to form multiple distinct populations of granules some of which associate with the IMC. A.** Co-immunofluorescence of ADAD2 and RNF17 in adult wildtype testes across selected pachytene spermatocyte developmental stages demonstrating colocalization. Roman numerals–testis tubule cross-section stage (VIII containing mid-stage pachytene spermatocytes, IX through XI containing late-stage pachytene spermatocytes). Asterisks—small granules and arrowheads—large granules. Red—RNF17, green—ADAD2, and blue—DAPI. Co-immunofluorescence of **B.** NANOS1 and ADAD2 or RNF17 demonstrating NANOS1 co-localization in both large and small ADAD2 or RNF17 granules (red—NANOS1, green—ADAD2 or RNF17, and blue–DAPI), **C.** PIWIL1 with ADAD2 or RNF17 in adult wildtype testes demonstrating co-localization in a subset of ADAD2 or RNF17 granules (red—PIWIL2 or PIWIL1, green—ADAD2 or RNF17, and blue–DAPI), and **D.** the mitochondrial marker COX IV and ADAD2 or RNF17 demonstrating a subset of small granules colocalize with the mitochondria but large granules do not (red–COX IV, green–ADAD2 or RNF17, and blue–DAPI). Asterisks—small ADAD2 or RNF17 granules. Arrowheads—large ADAD2 or RNF17 granules. 630x magnification for all images.

further confirmed in *Adad2* and *Rnf17* mutant testes (S4A Fig). This analysis demonstrated ADAD2 and RNF17 localize to the same granules in spermatocytes. Given this, further reference will be to the large or small ADAD2-RNF17 granules.

The mammalian spermatocyte contains five distinct germ cell granules [3,25], as identified by electron microscopy (EM). These granules are crucial for proper germ cell development

[5,6,33], though their exact protein composition and function are not wholly described [23]. In spite of this, several proteins are observed across four of the five granules, including DDX4 [20] and DDX25 [20,21]. The only granule known to be negative for DDX4 and DDX25 is referred to as the "cluster of 30 nm particles", which is observed via EM from late pachytene until the end of meiosis [3,25]. Previous analysis of ADAD2 granules has demonstrated ADAD2 does not colocalize with DDX25 [24], suggesting the ADAD2 granule is unique among the protein-associated granules. However, ADAD2's localization with DDX4 and the localization of RNF17 in relation to these markers is entirely undescribed.

To determine whether the large ADAD2-RNF17 granule could represent the protein-orphan "cluster of 30 nm particles" or if ADAD2 and RNF17 instead localize with one of the better described granules, we examined ADAD2 and RNF17 granule co-localization with DDX4 and DDX25 (S4B and S4C Fig). For both ADAD2 and RNF17, no co-localization with DDX4 was observed in large or small granules. In contrast, a subset of small ADAD2 granules as well as a similar subset of small RNF17 granules were observed to colocalize with DDX25 while large granules did not. To further define the molecular composition of the ADAD2-RNF17 granule, we examined the localization of a third granule associated RBP, NANOS1 [22,25] (Fig 4B). Although NANOS1 has been reported by immuno-EM to localize to a similar set of granules as DDX4 and DDX25, NANOS1 detection by this method was shown to be extremely weak [25] and thus may not be entirely representative of true NANOS1 localization. Supporting this notion, both ADAD2 and RNF17 large and small granules were positive for NANOS1. Together, the unique molecular signature of the large ADAD2-RNF17 granules defined here (negative for both DDX4 and DDX25 but positive for NANOS1) indicates they are not among the four described spermatocyte germ cell granules and thus, based on protein composition, represent a previously uncharacterized granule, possibly the "cluster of 30 nm particles".

Based on the apparent differences in DDX25 between the large and small ADAD2-RNF17 granules, we set out to determine whether there were other molecular differences between the two populations. Thus, we assessed the localization of two well defined granule proteins in relation to ADAD2 and RNF17. These proteins, PIWIL1 and PIWIL2, are both associated with processing of small non-coding RNAs known as piRNAs (4,13,14) as well as localizing to granule structures in the pachytene spermatocytes [14,20]. To date, both PIWIL1 and PIWIL2 are most closely associated with large ribonuclear complexes referred to as piRNA-p-bodies [34] which can be visualized in the spermatocyte cytoplasm and overlap in large part with the well-defined IMC granule [17]. RNF17 has been implicated as a major regulator of piRNA biogenesis via interaction with PIWIL1 [35] suggesting at least a subpopulation of RNF17 granules may also contain either PIWIL1 or PIWIL2. Thus, co-immunofluorescence was used to determine if ADAD2 and/or RNF17 localize with either PIWIL1 (Fig 4C) or PIWIL2 (S5A Fig) in the context of either the large or the small ADAD2-RNF17 granule. For both ADAD2 and RNF17, co-localization with PIWIL1 was observed in some, but not all, of the small granules. Additionally, a subset of the small ADAD2 and RNF17 granules were often found in close association, but not directly colocalized with, PIWIL2 granules. Despite their close localization, loss of neither ADAD2 nor RNF17 influenced the localization of PIWIL1 or PIWIL2 (S5B Fig). Given the lack of DDX4 in the small ADAD2-RNF17 granules, their occasional colocalization with PIWIL1 and proximity to PIWIL2-rich structures suggest at least some of the small ADAD2-RNF17 granules represent a previously unappreciated subpopulation of piRNA-associated granules.

Together, these observations demonstrate the ADAD2-RNF17 granules represent at least two distinct populations: a pool of small, heterogeneous granules and a molecularly distinct larger population. Given their molecular makeup, a portion of the small granules likely

represent a previously undefined subpopulation of piRNA-associated granules while the large granules are molecularly distinct and have a profile most similar to the cluster of 30 nm particles.

## Small ADAD2-RNF17 granules are a mix of IMC-associated and cytoplasmic aggregates

Association with PIWIL1 implies the small ADAD2-RNF17 granules may be associated with piRNA-based events, many of which occur near and around the mitochondria in an RNA granule-like aggregation referred to as intermitochondrial cement (IMC) [2,36]. Thus, we examined the co-localization of both ADAD2 and RNF17 with COX IV, a mitochondrial membrane protein [37]. Initial analysis demonstrated the co-localization profile of ADAD2 or RNF17 with COX IV was complex (Fig 4D). COX IV was rarely observed near or within the large ADAD2 or RNF17 granules. However, two populations of small ADAD2 and RNF17 granules were observed, some closely associated with COX IV signal and some not. This observation further supports the notion that the small ADAD2-RNF17 granule is distinct from the large and that there are at least two populations of small ADAD2-RNF17 granules, some as cytoplasmic aggregations not associated with the mitochondria and others part of or closely associated with the piRNA-associated IMC.

## The molecular composition and organelle association of the ADAD2-RNF17 granule changes across germ cell development

Initial analyses of ADAD2-RNF17 granules suggested they appeared sequentially, with small granules observed first followed by the large. Additionally, the small granule population was composed of at least two different subtypes, with distinct molecular compositions and localizations. These observations led us to wonder if the observed granule types represented different maturation steps linked to the developmental state of the cell. To examine this possibility, we quantified ADAD2 granule size in spermatocytes as a function of tubule cross-section stage (S6A Fig). This analysis revealed that small granules represented the majority of the population when ADAD2 is first detected, however they then declined in number rapidly until reaching a low in stage IX. Conversely, while a small number of large granules appeared with first detection of ADAD2, they dramatically accumulated over time reaching a maximum in IX and staying constant thereafter. An unexpected observation from this analysis was the detection of a third population of granules intermediate in size that stayed relatively constant throughout development. Together, these observations suggest the granule population transitions from primarily small granules to a mix of small and large which is thereafter maintained, a pattern suggestive of a population undergoing maturation from small to larger aggregates.

We next asked whether the developmental shift in granule size was mirrored in the granules' molecular composition. As our previous analysis had suggested heterogeneity in PIWIL1 colocalization with ADAD2, we measured ADAD2 and PIWIL1 colocalization as a function of stage (S6B Fig). As expected, a fraction of the small granules colocalized with PIWIL1 while nearly no large granules did, independent of stage. On the other hand, the intermediate granules, which initially colocalized with PIWIL1 at a level similar to the small granules, shifted across development to a profile much more similar to the large granule suggesting they may represent a transition state from the small to the large granule. We performed similar analyses comparing COX IV and ADAD2 localization (S6C Fig) which demonstrated a graduate transition away from COX IV association in the small granules while the large granules displayed limited COX IV localization throughout development. In contrast to both, the intermediate granules were found to have relatively low association early in development followed by a

rapid increase and then a graduate decline. Overall, these analyses support the notion that the ADAD2-RNF17 granule population represents a continuum from mitochondrially-associated, PIWIL1-containing, small granules which are likely a component of the IMC to mitochondria free, PIWIL1-negative, large granules of unknown function.

## The large ADAD2-RNF17 granule is a P-body that requires both proteins to form

Given the observation that the ADAD2-RNF17 granules represent a continuum, with small granules having a molecular signature indicative of piRNA processing, we next focused on characterizing the large granule. Previous reports had suggested that ADAD2 may be involved in translation regulation [26]. This, along with the clear localization of NANOS1 to the large ADAD2-RNF17 granule, suggested it may be a form of P-body, RNA granules classically associated with a wide range of RNA processes including translation [34,38–40]. In many cell-types P-bodies are marked by EDC3 [41,42]. Thus, we asked whether EDC3 also localized with ADAD2 (Fig 5A). This analysis demonstrated heterogeneity in the small granule population in mid-stage pachytenes with some granules EDC3-only positive, some ADAD2-only positive, and some double positive. Although small EDC3 granules were dramatically reduced later in development, those few remaining were generally ADAD2 positive as well. In contrast, large granules were exclusively double positive regardless of developmental stage.

Given EDC3 clearly marks the large ADAD2-RNF17 granule, we asked whether EDC3 required either ADAD2 or RNF17 to form large granules (Fig 5B). In wildtype testes, both large and small EDC3 granules are observed at stage IX and they are generally ADAD2 positive. In contrast, loss of either ADAD2 or RNF17 appeared to reduce or eliminate the large EDC3 granule population at this stage. The impact was even more dramatic later in development, where *Adad2* mutant spermatocytes formed no EDC3 granules while *Rnf17* mutant spermatocytes formed small, but not large, EDC3 granules. Although these analyses cannot eliminate the possibility that a protein aggregate of similar ultrastructure exists in the absence of ADAD2 or RNF17, the protein composition of that aggregate would differ drastically from the large ADAD2-RNF17 granule. Together, the evidence suggests the ADAD2-RNF17 granule is a large P-body like granule and its formation is dependent on both ADAD2 and RNF17.

## The large ADAD2-RNF17 granule has a unique structure with distinct domains

Initial localization studies of the large ADAD2-RNF17 granule suggested it may be roughly spherical. However, several unusual features were observed (see Fig 4B and 4D, RNF17 signal, for examples) including regions of low or no signal in the center of the granule, which suggested the granule may be composed of subdomains, similar to those observed in other granule types such as stress granules [38,43]. To better define the large ADAD2-RNF17 granule structure, we first examined ADAD2 localization to determine whether it displayed variable localization within the granule (Fig 6A). This preliminary analysis revealed ADAD2 forms a distinct funnel shape. The diameter of this structure can be seen as a ring, with an intense ADAD2 signal around the outer edge and a significantly weaker ADAD2 signal in the interior. These rings measure 1.036 μM ± 0.248 μM, (n = 24) across with the weakly positive interior region measuring 452.563 ± 129.085 nm, (n = 24). To eliminate the possibility of technical artifacts, imaging was repeated with an alternate antibody against ADAD2 ("93Term") [24], which requires an alternate antigen retrieval method. Further, to eliminate the possibility of incomplete antibody penetration, analyses were repeated on thick wildtype slides to ensure capture of the entire structure (S1 Movie). In all cases, ADAD2 localization appeared similar

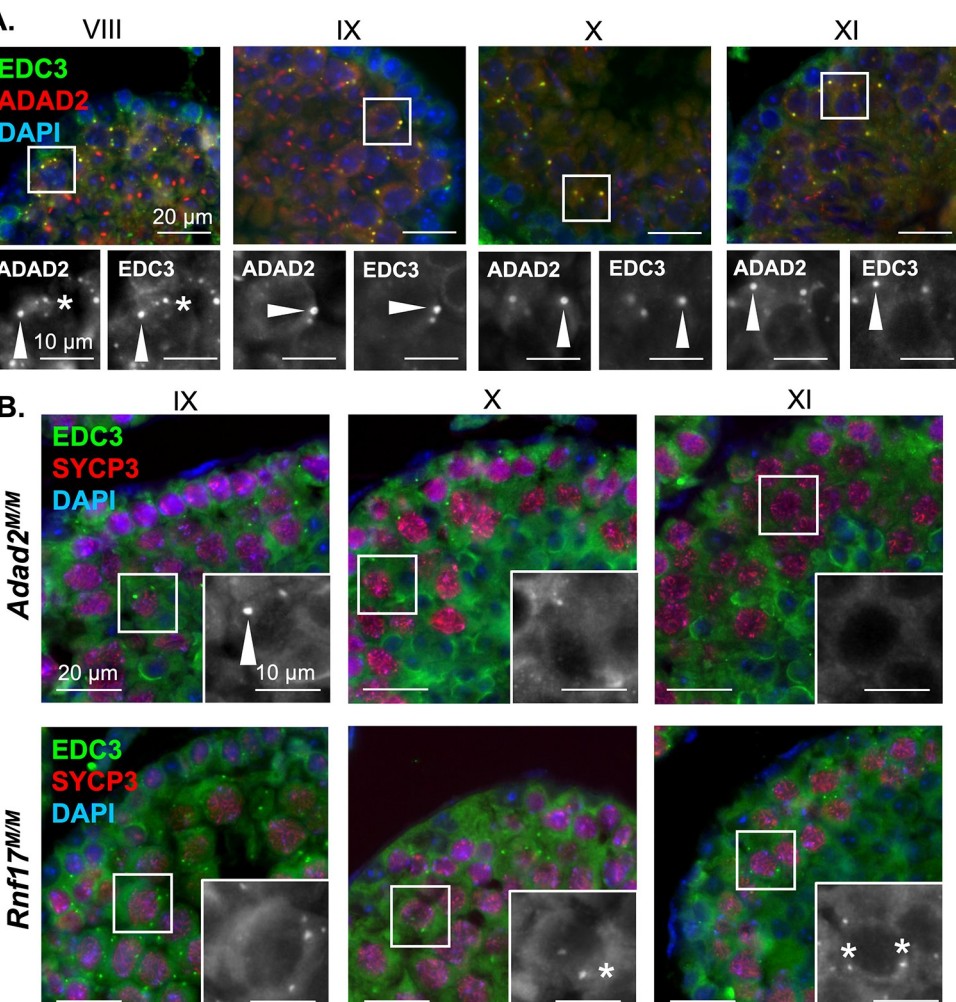

**Fig 5. The large ADAD2-RNF17 granule is a P-body. A.** Co-immunofluorescence of ADAD2 and EDC3 in adult wildtype testes across selected pachytene spermatocyte developmental stages demonstrating colocalization. Roman numerals–testis tubule cross-section stage (see Fig 4 legend for germ cell descriptions). Red–ADAD2, green–EDC3, and blue—DAPI. **B.** Immunofluorescence of EDC3 in $Adad2^{M/M}$ and $Rnf17^{M/M}$ mutant testes across selected spermatocyte developmental stages, marked by SYCP3, demonstrating loss of large EDC3 granules in both genotypes and aberrant formation of small EDC3 granules in $Rnf17^{M/M}$ mutants. Red–SYCP3, green–EDC3, and blue—DAPI. Inset–EDC3 alone. Asterisks—small granules and arrowheads—large granules. 400x magnification for all images.

suggesting the overall shape of the granule is cup or funnel shaped with a notably sharp and flat rim. Parallel analyses were performed for RNF17 which, much like ADAD2, displayed a similar localization pattern. However, measurements of the RNF17 granule diameter demonstrated it to be somewhat larger than ADAD2 at 1.412 µM ± 0.591 µM, (n = 26). Likewise, the interior measurements of the RNF17 ring (558.461 ± 285.590 nm, (n = 26)) suggested the overall RNF17-dense region of the granule to be slightly larger than the ADAD2-dense region.

To shed light on whether ADAD2 and RNF17 comprise different protein domains within the granule, co-localization studies using high-resolution confocal were performed (Fig 6B, S2 Movie). This analysis demonstrated distinct and differential localization of ADAD2 and RNF17 within the granule, with ADAD2 observed as a ring of dense aggregation around the outer rim and much weaker signal in the interior of the funnel. RNF17 was observed towards the exterior of the ADAD2 aggregation with moderate overlapping of the two domains.

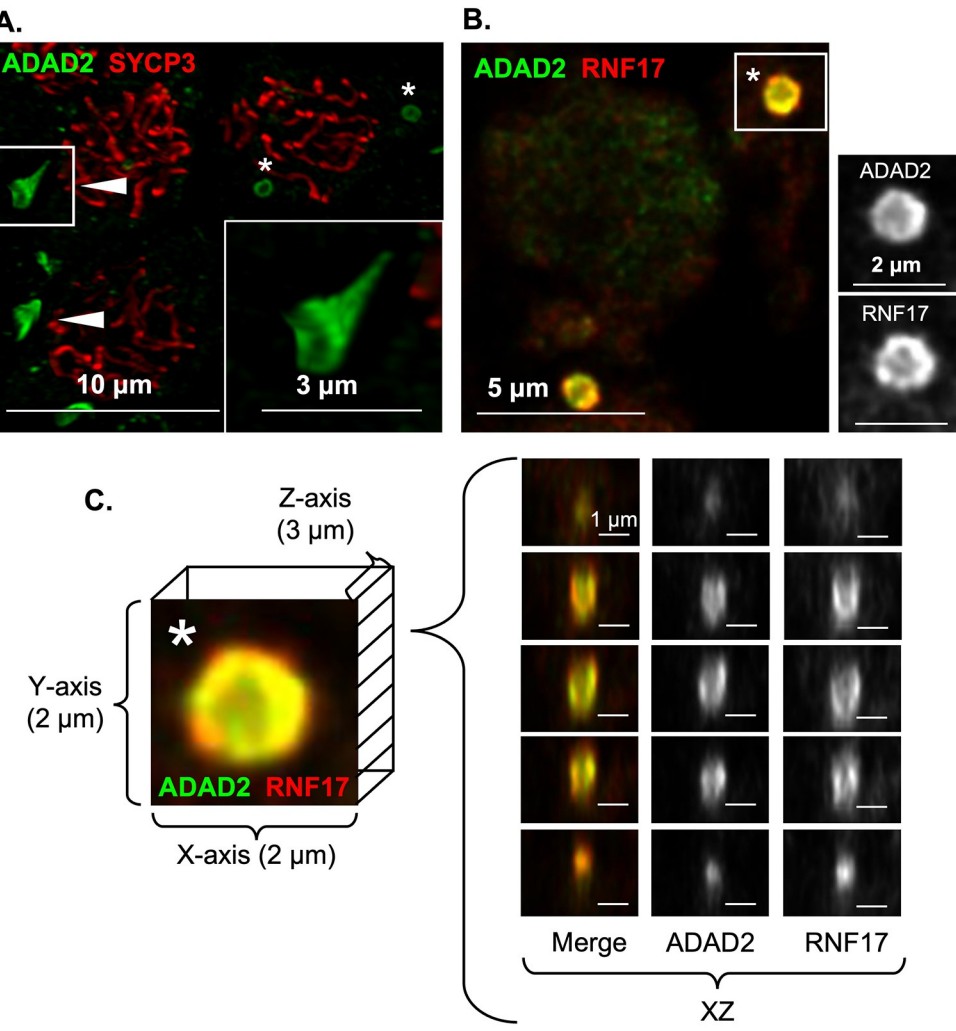

**Fig 6. ADAD2 and RNF17 form a uniquely shaped granule.** High-resolution confocal of **A.** ADAD2 in late pachytene spermatocytes counterstained with SYCP3 demonstrating a distinct funnel shaped structure. Red–SYCP3, green–ADAD2. 1000x magnification. **B.** Co-localization of ADAD2 and RNF17 in wildtype testis sections. Red—RNF17, green—ADAD2, and blue–DAPI. 1000x magnification. **C.** Selected ADAD2-RNF17 granule (asterisk in B) with representative XZ planes demonstrating the interior of the funnel shaped structure and the relative localization of ADAD2 and RNF17. Red—RNF17, green—ADAD2, and blue—DAPI. 1000x magnification.

Further, RNF17 appeared to be even less enriched in the interior of the funnel than ADAD2. This localization pattern held throughout the 3D structure of the granule (Fig 6C), with the RNF17 domain observed exterior to, but retaining the same shape of, the ADAD2 domain. Together, these findings demonstrate that the ADAD2-RNF17 granule has distinct regions, and these regions are defined by enrichment of either ADAD2 or RNF17.

### Translation regulators localize to distinct regions of the large ADAD2-RNF17 granule

Given ADAD2 and RNF17 show distinct regions of enrichment within the larger granule we wondered whether NANOS1 did the same. Confocal localization of NANOS1 relative to both ADAD2 and RNF17 (Fig 7A) demonstrated a distinct NANOS1 region of the granule that colocalizes primarily with the ADAD2 domain and resides almost entirely inside of the RNF17

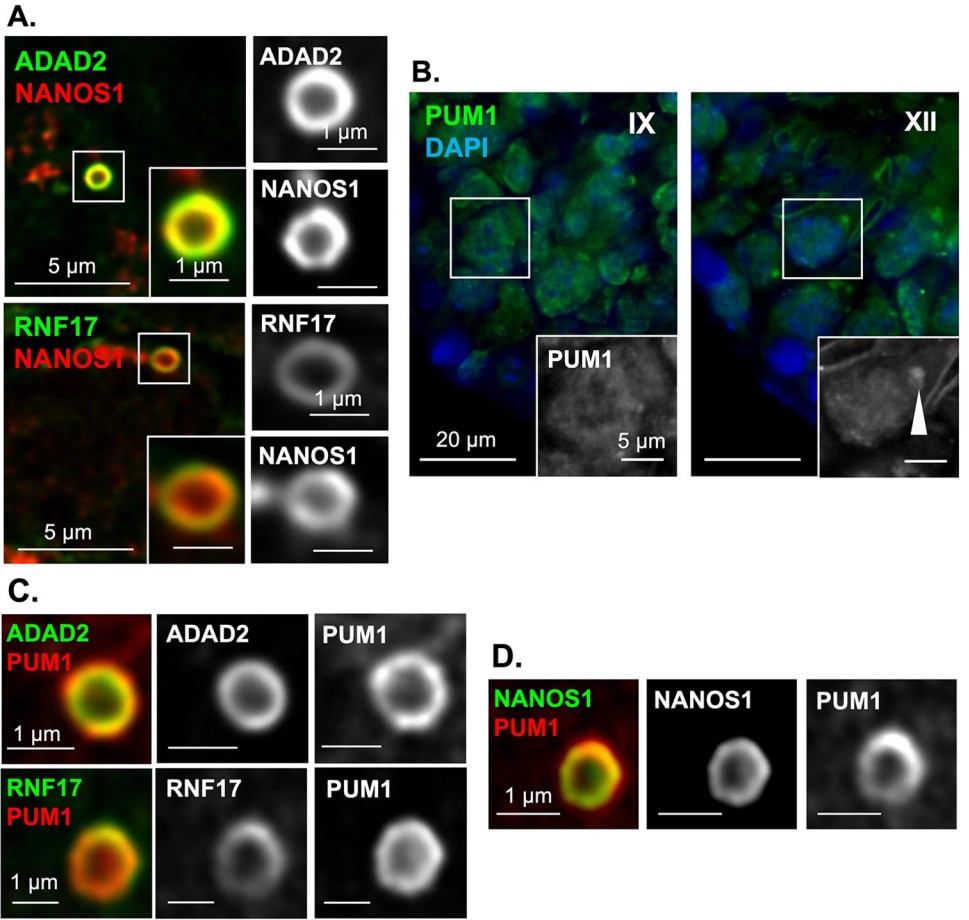

**Fig 7. NANOS1 and PUM1 reside in distinct domains of the large ADAD2-RNF17 granule. A.** Confocal imaging of ADAD2 and RNF17 with NANOS1 showing colocalization of NANOS1 to the ADAD2-RNF17 granule as well as differential localization relative to ADAD2 and RNF17. Red–NANOS1, green–ADAD2 or RNF17. 1000x magnification. **B.** PUM1 immunofluorescence in adult wildtype testes demonstrating aggregate accumulation (arrowhead) late in spermatocyte development. Roman numerals–testis tubule cross-section stage. 400x magnification. Confocal imaging of **C.** PUM1 with ADAD2 and RNF17 (red–PUM1 and green–ADAD2 or RNF17) and **D.** NANOS1 and PUM1 (green–NANOS1 and red–PUM1) demonstrating distinct granule domains. 1000x magnification.

domain. We next examined PUM1, a well described NANOS1 interacting partner [44,45], that has been implicated in translation regulation in multiple systems [46–48]. Given ADAD2's apparent role in translation regulation and the close association of ADAD2 with NANOS1, we wondered whether PUM1 may also localize to the ADAD2 granule and facilitate ADAD2's function. Previous reports have demonstrated PUM1 is expressed in spermatogonia and spermatocytes and has no detectible regions of aggregation when assessed by standard fluorescent microscopy [46], However, our analysis (Fig 7B) demonstrated relatively large PUM1 aggregates in late-stage spermatocytes. This localization pattern very closely mirrors that of the large ADAD2-RNF17 granule. As such, we leveraged confocal microscopy to more carefully examine PUM1 localization relative to ADAD2 (Fig 7C). Much like NANOS1, a portion of the PUM1 signal overlapped with ADAD2 and fell just inside the RNF17 domain of the ADAD2-RNF17 granule suggesting it likely overlapped with NANOS1. Examination of NANOS1 and PUM1 via confocal demonstrated this to be the case (Fig 7D). Together, these analyses confirm the localization of multiple translation regulators to the ADAD2-RNF17 granule and demonstrate them to have specific domains within the granule.

## The large ADAD2-RNF17 granule is associated with the endoplasmic reticulum

No other mammalian germ cell granule is known to have the distinct shape of the large ADAD2-RNF17 granule. As germ cell RNA granules are not bound by membranes of their own [3], they are shaped by their protein interactions and the cellular environment. The non-spherical shape of the ADAD2-RNF17 granule suggests contact with another cellular component. We therefore sought to determine if the large ADAD2-RNF17 granule associates with a membrane bound organelle which may provide the surface to shape the ADAD2-RNF17 granule. In C. elegans specific RNA granules are tightly associated with the nuclear membrane, which influences their shape [49–51]. Additionally, P-bodies have recently been shown to associate with the endoplasmic reticulum (ER) which modulates their formation and behavior [52]. Thus, we examined the co-localization of both ADAD2 and RNF17 with markers of the nuclear membrane (Lamin A/C) [53] and the ER (SERCA1) [54]. This analysis demonstrated Lamin was notably excluded from and generally distally located to sites enriched for either ADAD2 or RNF17 (S7A Fig)

In contrast, SERCA1 was not excluded from large granules of either ADAD2 or RNF17 and occasional regions of SERCA1 and ADAD2 or RNF17 co-enrichment were detected (S7B Fig) suggesting potential association of the ER with the large ADAD2-RNF17 granule. To better define the spatial association of the ER with the ADAD2-RNF17 granule, we examined the co-localization of ADAD2 and SERCA1 using confocal microscopy (Fig 8A). These analyses clearly identified the ADAD2-RNF17 granule along with defining the tubular structure indicative of the ER [55]. Examination of the large ADAD2-RNF17 granule demonstrated a distinct pattern of association between SERCA1 and ADAD2 (Fig 8B) with SERCA1 observed along the interior of the ADAD2-RNF17 funnel shape with no substantial signal occupying the center portion of the structure.

SERCA1 resides in the membrane of the ER [56], thus its association with ADAD2 does not definitively demonstrate association with an intact ER tubule. In order to conclusively demonstrate ER association, we further examined the colocalization of ADAD2 with an ER lumen resident protein, protein disulfide isomerase (PDI). Under normal conditions, PDI facilitates protein folding [57,58] and it has been shown to form a granule-like structure in rat spermatocytes [59]. As for SERCA1, immunofluorescent detection of PDI with either ADAD2 or RNF17 demonstrated frequent colocalization by standard microscopy (S7C Fig). A more detailed analysis showed that while small granules occasionally were enriched for PDI, large granules showed consistent colocalization of PDI and ADAD2 or RNF17, confirming the large granule associated with the ER and in particular, with one of the well-defined protein folding chaperones resident therein. Further analysis by confocal microscopy (Fig 8C) demonstrated that, like SERCA1, PDI resided internal to ADAD2 but did not occupy the entire volume of the funnel. Together, these results suggest the unique shape of the ADAD2-RNF17 granule results from its association with the endoplasmic reticulum, though the driving factors behind this association remain unclear.

## Loss of both ADAD2 and RNF17 phenocopies the *Adad2* and *Rnf17* phenotype

To assess whether the loss of both ADAD2 and RNF17 had a more profound impact on germ cell development than singular loss, $Adad2^{M/M}$:$Rnf17^{M/M}$ mice were generated. Dual loss of both ADAD2 and RNF17 was confirmed via Western blot (S8A Fig). Though female $Adad2^{M/M}$:$Rnf17^{M/M}$ (Double mutant, DM) mice were fertile, males were completely infertile and demonstrated post-meiotic cell loss (S8B Fig). When round spermatids were assessed in DM males, a

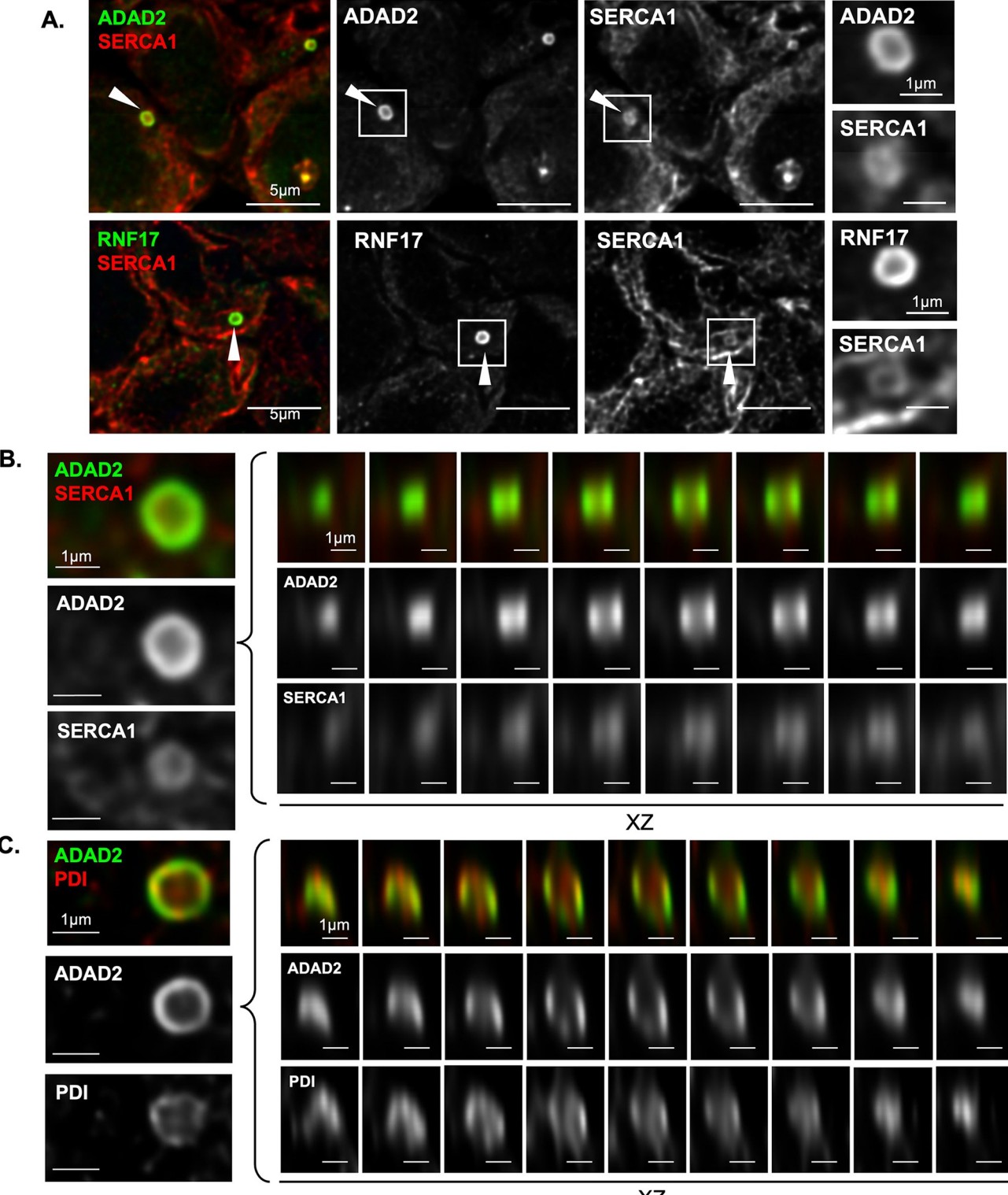

**Fig 8. The large ADAD2-RNF17 granule associates with the endoplasmic reticulum. A.** Confocal imaging of the ER-marker SERCA1 and ADAD2 or RNF17 in adult wildtype testes showing SERCA1-enriched regions associate with the interior region of the large granules. Red–SERCA1, green–ADAD2 or RNF17. 1000x magnification. XY and XZ views of **B.** ADAD2 and SERCA1 and **C.** ADAD2 and PDI within a large granule. Red–SERCA1 or PDI, green–ADAD2. 1000x magnification.

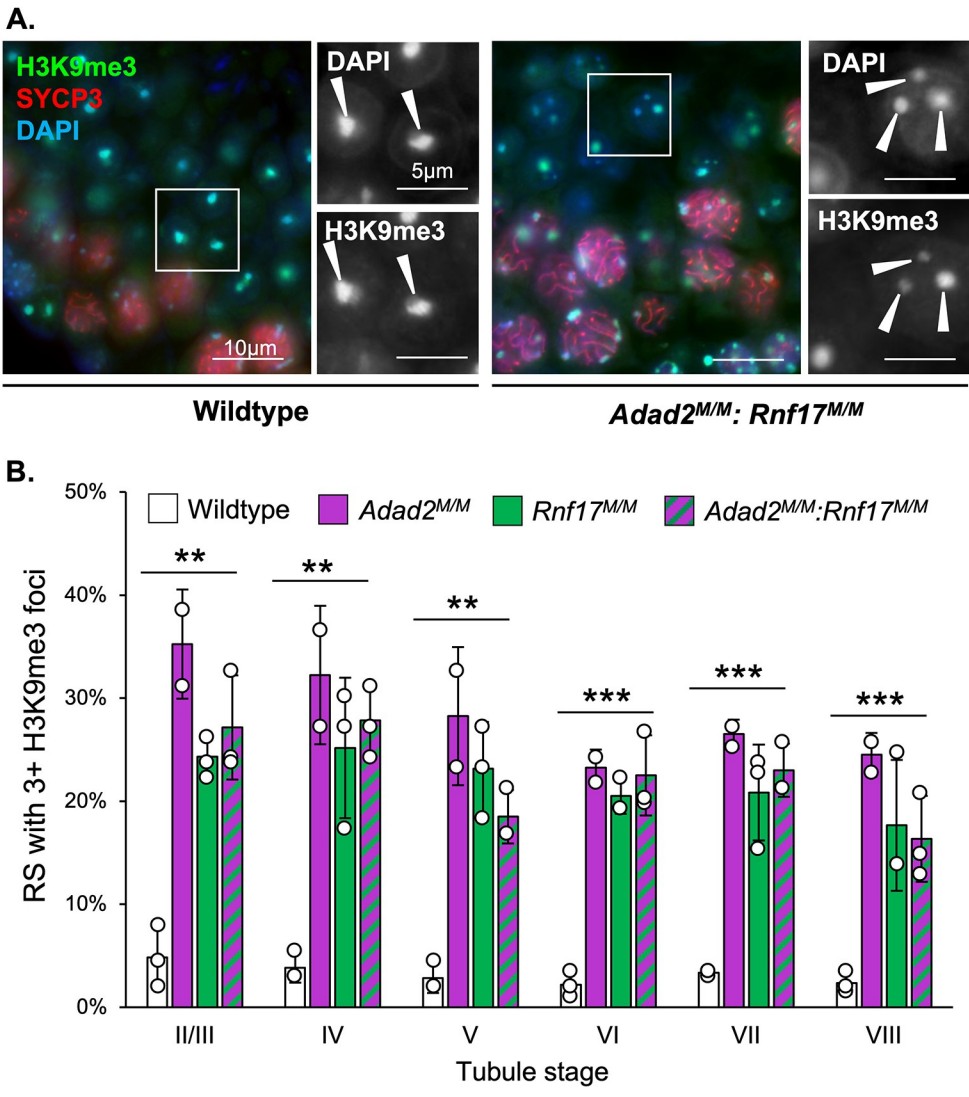

**Fig 9. *Adad2:Rnf17* double mutants have a very similar chromocenter phenotype to that observed in single mutants. A.** Immunofluorescence of the heterochromatin mark H3K9me3 along with DAPI staining in adult wildtype and $Adad2^{M/M}$: $Rnf17^{M/M}$ testes. Red—SYCP3, green—H3K9me3, and blue—DAPI. Arrowheads–DAPI or H3K9me3 foci in round spermatids. 400x magnification. **B.** Quantification of round spermatids (RS) with 3 or more chromocenter-like structures as marked by H3K9me3 staining per developmental stage in adult wildtype, $Adad2^{M/M}$: $Rnf17^{M/M}$, $Rnf17^{M/M}$ (n = 3), and $Adad2^{M/M}$ (n = 2) testes demonstrating a significant increase in $Adad2^{M/M}$: $Rnf17^{M/M}$ round spermatids relative to wildtype. Data are mean ± s.d. Significance was calculated using an unpaired, one-tailed Student's t-test between wildtype and $Adad2^{M/M}$: $Rnf17^{M/M}$ (*$P < 0.05$, **$P < 0.005$, ***$P < 0.0005$).

significant decrease was observed compared to wildtype (S8C Fig), similar to either ADAD2 or RNF17 loss. Further, examination of the heterochromatin landscape in DM round spermatids demonstrated heterochromatin abnormalities consistent with those observed in both $Adad2^{M/M}$ and $Rnf17^{M/M}$ single mutants (Fig 9A). Quantitation of these abnormalities ultimately revealed that the double mutant males exhibited similar frequencies of abnormal H3K9me3 foci as observed in both single mutants (Fig 9B). As the phenotype of the double mutants mimics that of either single mutant, these results suggest that the formation of the ADAD2-RNF17 granules, or the proteins' interaction, is more crucial to successful germ cell development than the presence of the proteins themselves.

## Discussion

In mammalian meiotic male germ cells, germ cell granules are important sites of RNA metabolism required for their normal differentiation. Traditionally, spermatocytes are thought to contain five morphologically distinct granule types and loss of many granule components leads to male infertility. In spite of this, there is limited knowledge regarding the protein composition of, function of, and relationship between these granules. To that end, this work aimed to characterize a recently identified granule that contains the RNA binding protein ADAD2, which is required for male fertility [24]. After identifying a second RNA binding protein, RNF17 [8], as an ADAD2 interacting partner, genetic knockout models combined with localization studies demonstrated ADAD2 and RNF17 are co-dependent on one another to form multiple distinct populations of ADAD2-RNF17 granules. Protein composition studies further showed ADAD2-RNF17 granules are comprised of at least three molecularly distinct subpopulations, one of which is associated with the well-known piRNA-associated IMC. Analysis of granule composition and organelle association across spermatocyte development demonstrated development-dependent transitions from one granule type to another. These transitions ultimately gave rise to a large, ER-associated granule containing numerous translation regulators. Lastly, dual loss of both ADAD2 and RNF17 via genetic ablation suggested loss of the granules or the ADAD2-RNF17 interaction, as opposed to the individual proteins, is the primary driver of the *Adad2* and *Rnf17* mutant phenotypes. Together, these studies have genetically defined the importance of multiple novel germ cell granules in male fertility and have additionally revealed new aspects of germ cell granule biology that establish future approaches for the study of other germ cell granules.

Although nearly all granule studies discern differences in granule presence as a function of germ cell type (for example spermatogonia versus spermatocyte) [2,60–63], detailed molecular analyses of a single granule type across a narrow range of germ cell development, as done here, are rare. As a result, we know very little about whether granules transition from one type to another. Our analyses of ADAD2-RNF17 granule size, molecular composition, and organelle association by cell developmental stage are an initial attempt to address this question. Overall, our data suggest a model of granule maturation (Fig 10) wherein small granules, cytoplasmic or mitochondrially associated, coalesce to form transient intermediate granules which then shed PIWIL1, acquire PUM1, and finally mature into an ER-associated large granule. It should be noted, this is only one of several maturation trajectories consistent with the observations reported herein. One clear caveat of the above analysis is that it does not allow direct observation of granule maturation across germ cell development. Direct observation of these highly dynamic events will require live imaging of developing spermatocytes expressing labeled proteins. Fortunately, promising approaches have been developed in other systems, such as stress granule formation [43,64,65], and these represent an exciting future avenue for germ cell granule research.

Our work suggests a population of granules undergo maturation, or at the least extensively share components. This finding brings into question the idea that the individual granule types are distinct from one another. Rather, the identification of distinct ADAD2-RNF17 granule pools suggests that as opposed to distinct units, the different granule types may instead represent an interconnected network of RNA biogenesis or regulation sites. Several previous observations support this hypothesis as well. First, there are multiple proteins observed across many granules [23]. As an example, NANOS1 appears to be shared across all ADAD2-RNF17 granule subtypes as well as between the large ADAD2-RNF17 granule and the other traditionally defined granules. Second, at least two granules have already been shown to actively share components [3,20,25,66]. This may also to be the case for the ADAD2-RNF17 granules. For

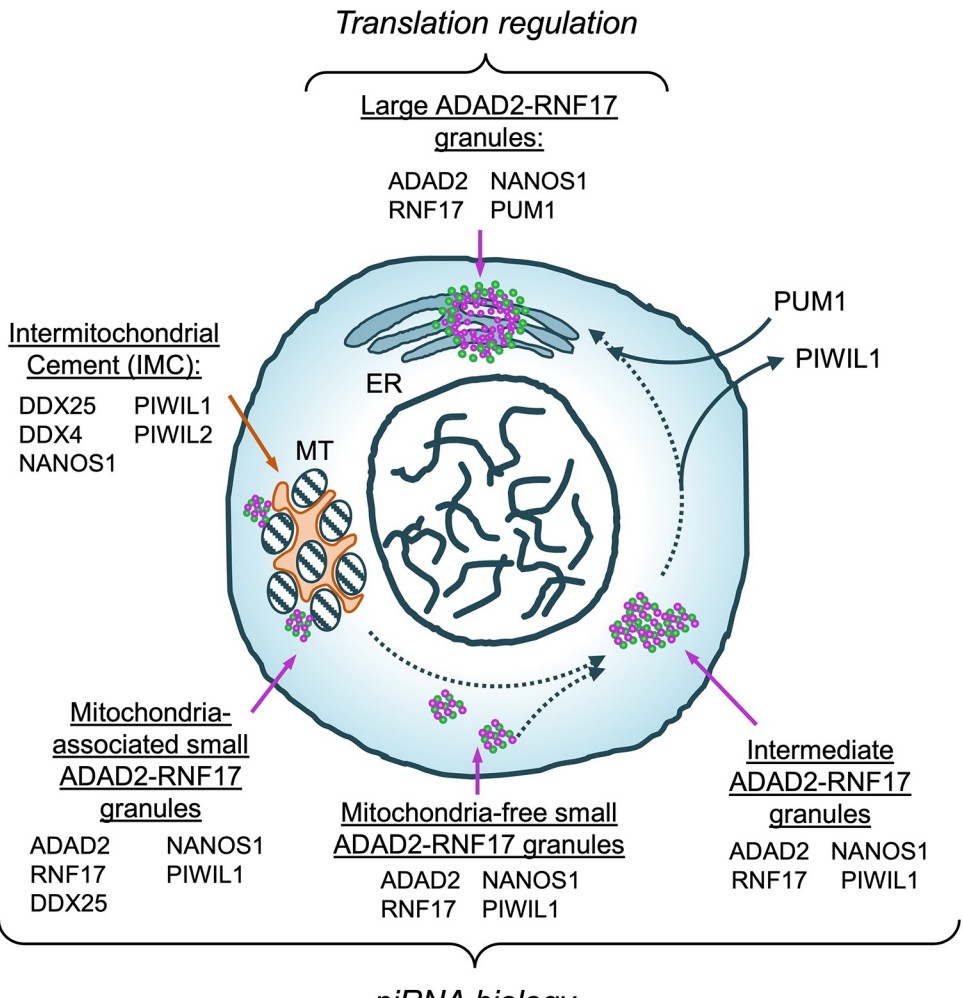

**Fig 10. Proposed model of ADAD2-RNF17 granule maturation.** Morphologically and molecularly distinct ADAD2-RNF17 granules (green and purple) as well as the IMC (orange) shown. Protein components of each granule listed. Dashed arrows indicate potential avenues of maturation. Solid arrows represent protein accumulation or loss. Potential functions for each granule population indicated in italics. MT–mitochondria, ER–endoplasmic reticulum.

example, the small ADAD2-RNF17 granules are DDX4-negative but DDX25-positive. Previous reports have suggested the only RNA granule in spermatocytes that is DDX25-positive is the chromatoid body (CB) precursor [67], which is also DDX4-positive [66]. As such, it may be that the small ADAD2-RNF17 granules are remodeled and form part of the CB precursor. Likewise, it may be that the large ADAD2-RNF17 granules share components with the CB precursor.

Important to this question, it seems likely that not all granules or their intermediates will be detectable by the traditional approach of electron microscopy (EM). For example, of the ADAD2-RNF17 granule populations examined here, only the large granule would be detected by EM based on its size and density [3]. However, modern imaging techniques have progressed dramatically in the intervening years [23]. As a result, high- or super-resolution microscopy is likely to be a preferred, more powerful, and more accessible tool to address the question of granule relationships.

That we observe distinct populations of ADAD2-RNF17 granules in a developmentally regulated manner begs the question of whether they have distinct functions. Based on protein composition and mitochondrial association of the small granules it seems likely they function in some aspect of piRNA biology. Supporting this notion, two recent reports have demonstrated ADAD2 influences piRNA biogenesis in a manner very similar to RNF17, likely via the interaction between the two proteins [68,69]. However, the IMC is the only definitively known site for piRNA processing in meiotic male germ cells [14], yet a subset of small ADAD2-RNF17 granules are clearly separate from mitochondria, demonstrating them to be independent of the IMC. These observations make it tempting to postulate the two different types of small ADAD2-RNF17 granules represent sites of different piRNA processing or action steps. Future efforts will focus on dissecting the exact functions of the mitochondria-associated and mitochondria-free ADAD2-RNF17 granules.

While the small ADAD2-RNF17 granules are likely components of the piRNA biogenesis pathway, evidence reported herein suggests this may not be the case for the large granules, which have a different protein composition as well as organelle association. While it is possible the large granule also serves some PIWIL-protein independent function in the piRNA pathway, a more reasonable explanation is that the large ADAD2-RNF17 granule acts to regulate a different aspect of RNA biology, most likely translation. Several lines of evidence support this conclusion. First, previous reports have demonstrated ADAD2 loss results in altered translation of a specific set of transcripts, many of which are involved in heterochromatin remodeling [26]. This matches well with the observations reported herein where mutation of either *Adad2* or *Rnf17* results in dramatic changes to post-meiotic germ cell chromatin structure. Further, this phenotype is distinct from that observed for other piRNA processing mutants such as *Piwil1* and *Tdrd5* [62,63] suggesting it may be driven by a non-piRNA-associated function for both ADAD2 and RNF17. Additionally, the large granule contains translation regulators not observed in the small granule and these regulators appear to accrue during development. The *Drosophila melanogaster* homolog of NANOS1 is known to facilitate localized translation regulation in the embryo [70] while in *Xenopus*, Nanos1 facilitates translational repression in the germline [71]. Likewise, PUM1's primary function in other systems is as a translation regulator [44,46,72]. Lastly, the large ADAD2-RNF17 granule appears to have an intimate association with the endoplasmic reticulum. Although traditionally associated with translation of secreted proteins, recent evidence has revealed the ER regulates translation across much of the transcriptome [73]. As such, the ER is a nexus for translation control. Together, these observations argue that the large ADAD2-RNF17 granule may act as a site of translation regulation. Future efforts will focus on fully describing the exact molecular events occurring in the large granule.

In addition to revealing the multifunctional nature of the ADAD2-RNF17 interaction, the data presented here provides insight into general RNA granule biology. First, localization of ADAD2 within the granule identified distinct regions of high and low density, in particular along the rim of the structure and relative to the interior. This is reminiscent of what is observed in stress granules, especially for the stress granule protein G3BP1 [65,74,75]. Stress granules are well-described, RNA-rich, non-membrane bound structures that form under specific cellular stresses [40]. As such, they represent an excellent model system to better understand granule formation. Recent work has demonstrated that nucleation of G3BP1 drives stress granule formation, which is ultimately the result of liquid-liquid phase separation [64]. It is unknown whether this is a primary driver of ADAD2-RNF17 granule or the other germ cell granules' formation. However, it is known that DDX4 undergoes phase separation *in vitro* [76] and multiple groups have proposed phase separation to be the primary driver of granule formation *in vivo* [77–80].

Although it seems likely the ADAD2-RNF17 granule is formed, in part, by phase separation other observations suggest additional levels of regulation. In contrast to stress granules, which are spherical or oblong in nature [77], the ADAD2-RNF17 granule displays multiple facets that appear flat and has a distinctive "empty" core within which resides two markers of the ER, SERCA1 and PDI [56,81]. This suggests physical interaction with the ER membrane may generate some sort of physical or mechanical constraint on ADAD2-RNF17 granule formation, perhaps similar to the constraints mitochondrial tethering puts on IMC components [82,83]. ER tubules range from twenty to sixty nanometers [84], thus the geometry of the ER-ADAD2 granule interaction is not entirely straight forward. One possibility is that the granule wraps around a node of three-way junctions. In support of this, it was recently shown that interaction between stress granules and the three-way junctions of the ER network drove stress granule shape and fission [52]. Thus, it seems feasible that the structure or behavior of the ADAD2-RNF17 granule, like stress granules, may be regulated by the ER. Further, it points to a potentially conserved mechanism of regulation between stress granules and the large ADAD2-RNF17 granule.

In total, this work has identified several protein components of an entirely undescribed germ cell granule population. Importantly, this work lays the foundation to address multiple outstanding questions regarding the ADAD2-RNF17 granules. These include the exact molecular function of the granule subpopulations, the purpose of the protein subdomains within the large granule structure, and whether other associated proteins drive one or both of these. Study of the ADAD2-RNF17 granules has already provided novel insight into the biology of other germ cell granules, as such addressing these exciting questions should not only inform on ADAD2 and RNF17 biology but on germ cell granules as fundamental drivers of male germ cell differentiation.

## Materials and methods

### Ethics statement

All animal use protocols were approved by the Rutgers University animal care and use committees. Mouse procedures were conducted according to relevant national and international guidelines as outlined in the Guide for the Care and Use of Laboratory Animals and provisions set forth by the Animal Welfare Act. Adherence to these guidelines was overseen on the institutional level by the Rutgers Institutional Animal Care and Use Committee and all animal procedures were approved under Rutgers IACUC ID TR202000034.

### Animal care and model generation

Generation of $Adad2^{M/M}$ mice was as described by Snyder et al. [24] and $Rnf17^{M/M}$ mice were as described by Pan and colleagues [8]. $Adad2$:$Rnf17$ mice were generated by breeding $Adad2^{M/M}$ females to $Rnf17^{+/M}$ males as well as $Rnf17^{M/M}$ females to $Adad2^{+/M}$ males to generate $Adad2^{+/M}$: $Rnf17^{+/M}$ offspring. These were intercrossed to generate $Adad2^{M/M}$: $Rnf17^{M/M}$, $Adad2^{+/+}$: $Rnf17^{M/M}$, $Adad2^{M/M}$: $Rnf17^{+/+}$ and $Adad2^{+/+}$: $Rnf17^{+/+}$ experimental animals. All mice were housed in a sterile, climate-controlled facility on a 12 h light cycle. Mice were fed LabDiet 5058 irradiated rodent chow and had access to food and water ad libitum.

### Immunoprecipitation

Testes were collected from 42 dpp $Adad2^{M/M}$, $Rnf17^{M/M}$, and wild-type mice (n = 3) and flash frozen. Tissue was ground in liquid nitrogen and total protein extracted in RIPA buffer (50mM Tris-HCl (pH 8), 150 mM NaCl, 1% (v/v) NP-40, 0.5% (w/v) sodium deoxycholate,

0.1% (w/v) SDS) with protease inhibitors (Thermo Scientific, 1 tablet per 10 mL) at a ratio of 1 ml buffer to 100 mg tissue. Tissue was precleared with Protein A beads (Thermo-fisher) equilibrated in RIPA (10 μL beads per 100 mg tissue) for one hour at 4°C. For every 1 mL of lysate, 4 μL of either ADAD2 antibody (94 AP [24]) or 2.5 μg RNF17 antibody (Proteintech) was added. Samples were incubated with antibody overnight at 4°C with constant rotation. Subsequently, beads were added and left to bind for 2 hours at 4°C with rotation followed by bead washing with 0.25X TBS (5 mM Tris Base, 37.5 mM NaCl). For mass spectrometry, protein was extracted from beads with 30 μL glycine elution buffer (0.2 M glycine, pH 2.6) per 1 ml of lysate followed by neutralizing with Tris pH 8.0. Glycine elution was confirmed via SYPRO gel staining (as per manufacturer's protocol, see below) and western blotting (S9 Fig). Analysis of IP protein by mass spectrometry was performed as below. For western blot confirmation of identified interactions, IP beads were boiled for 5 minutes at 95°C in SB loading dye (30 μL dye per 1 mL). Eluent was then isolated from the beads and prepared for western blotting along with controls (input, flow-through, and first wash).

## Mass spectrometry

Protein eluents for IP-MS were digested with trypsin after resuspension in 20 mM Tris-HCl pH 8.0. Samples were reduced with DTT and alkylated with iodoacetamide prior to addition of trypsin for overnight digestion at 37°C followed by quenching with formic acid and salt removal by p10 ZipTip. Samples were analyzed at Northwestern University Proteomics on a ThermoFisher Orbitrap. Data was then analyzed by the Mass Spectrometry and Protein Chemistry Service at The Jackson Laboratory using MASCOT against the SwissProt 2015_08 Mus musculus database. Fixed modifications were set for carbamidomethyl and variable modifications for acetyl/phosphorylation. Peptide mass tolerances were set to 25 ppm and 0.2 Da for intact and fragments, respectively. Proteins of interest were identified by comparison of wild-type and mutant IP samples. Underlying data can be found in S1 Data.

## Protein isolation and Western blotting

Testes were collected from adult (60–70 dpp) and 21 dpp $Adad2^{M/M}$, $Rnf17^{M/M}$, and wild-type male mice and flash frozen. Tissue was ground in liquid nitrogen and total protein was extracted by RIPA buffer with protease inhibitors at a ratio of 1 ml buffer to 100 mg tissue. Protein concentration was determined via the DC protein assay (BioRad) as per manufacturer's instructions. Samples were diluted in RIPA buffer, and 1M dithiothreitol and loading dye (100 mM Tris-HCl (pH 6.8), 4% (w/v) SDS, 0.05% (w/v) bromophenol blue, 20% (w/v) glycerol) was added in a ratio of 2:1:2. Samples were boiled at 95°C for 5 minutes prior to loading.

For Western blotting of total protein, 20 μg protein per sample was electrophoresed on 10% acrylamide gels for ADAD2 detection and 50 μg protein per sample was electrophoresed on 6% acrylamide gels for detection with RNF17. For Western blotting of IP panels, 20 μL each of input, flow-through, and wash along with 20 μL of IP diluted 1:10 was loaded per well. Following wet transfer of proteins to a PVDF membrane (BioRad), membranes were blocked and incubated overnight with primary antibody at 4°C (anti-ADAD2 1:1000; anti-RNF17 (Proteintech) 1:1000). Blots were incubated in secondary antibody (Goat-Anti-Rabbit HRP, (Biorad) 1:2000) for one hour at room temperature. Images were developed with SuperSignal West Pico PLUS Chemiluminescent Substrate (Thermo Scientific) and visualized using an Azure Biosystems C600 imager. Equal loading and transfer was confirmed via SYPRO-Ruby staining (see below and S10 Fig).

## Western blot membrane and gel staining

Following visualization of blot, membranes were stained with SYPRO-Ruby Protein Gel Stain (Lonza Rockland) to confirm equal loading. Membranes were stained following manufacturer's instructions (Molecular Probes) and visualized using an Azure Biosystems C600 imager set to UV 302. For IP gel staining, gels were processed as per manufacturer's instructions after electrophoresis and imaged with the same method as for membranes.

## Hematoxylin and Eosin (H & E) staining

Testes from 60–70 dpp $Adad2^{+/M}$:$Rnf17^{+/+}$ and $Adad2^{M/M}$:$Rnf17^{M/M}$ males were collected and fixed overnight in Bouins Solution (Sigma Aldrich). Tissue was rinsed in deionized water, dehydrated in increasing concentrations of ethanol, embedded in paraffin wax and cut into 4 µm sections. Slides were deparaffinized in xylenes and rehydrated in decreasing concentrations of ethanol before staining with Harris Hematoxylin (Sigma Aldrich). Slides were rinsed in water and partially dehydrated before staining with Eosin Y (Sigma Aldrich). Slides were then dehydrated and mounted with Permount mounting medium (Sigma Aldrich). Samples were visualized on a custom-built microscope (Zeiss) with fluorescent and brightfield capabilities.

## Round spermatid quantification

Testes from adult wild-type, $Adad2^{M/M}$, $Rnf17^{M/M}$, $Adad2^{M/M}$:$Rnf17^{+/+}$, $Adad2^{+/+}$:$Rnf17^{M/M}$ and $Adad2^{M/M}$:$Rnf17^{M/M}$ (double mutant or DM) testes were collected and fixed overnight in 4% PFA. Tissue was rinsed in PBS, dehydrated in increasing concentrations of ethanol, embedded in paraffin wax and cut into 4 µm sections. Slides were deparaffinized in xylenes and rehydrated in decreasing concentrations of Ethanol before staining with DAPI Fluoromount-G (Southern Biotech).

Histological parameters previously described [85] and staging criteria (below) were used to quantify the number of round spermatids per tubule and number of round spermatid-containing tubules per sample. Round spermatid morphology was quantified in DAPI-stained samples. For total round spermatids, ten tubules per quantified stage were assessed per biological replicate (n = 3). For chromocenter analyses, spermatids were binned by number of intense H3K9me3-staining structures (1, 2, or 3+). Two hundred round spermatids per stage per biological replicate (n = 3) were counted. For both counts, totals and averages (means) for each genotype were calculated, as well as s.d. An unpaired, one-tailed Student's t-test was used to identify significant differences by genotype. Underlying data can be found in S1 Data.

## Immunofluorescence

Testes were dissected from adult mice and fixed overnight in 4% (w/v) PFA in PBS. Tissue was rinsed in PBS and dehydrated in increasing concentrations of ethanol before embedding in paraffin wax. Applications used 4 µM sections. Antigen retrieval was performed by boiling slides in Tris-EDTA pH 9.0 (10 mM Tris-HCl, 1 mM EDTA, and 0.05% Tween) on low power for 30 min or 15.6 mM Citrate pH 5.95, on high power for 2 minutes, medium power for 7 minutes and 20 minutes at room temperature (93-Term only [24]). Slides were incubated with primary antibodies overnight at room temperature, washed briefly, then incubated with appropriate fluorophore-labeled secondaries for one hour at room temperature, washed and then mounted. Antibodies and their conditions can be found in S2 Table. Slides were mounted using DAPI Fluoromount-G and stored at 4˚C with light protection.

Slides were visualized on a custom-built microscope (Zeiss) with fluorescent and bright-field capabilities. Each channel was imaged individually through MetaMorph imaging software (Molecular Devices) and color-combined using the built-in color combine tool. Provided images are representative of three or more biological samples. Signal intensity was matched across slides by matching background (interstitial) signal intensity. Developmental stages were determined according to the parameters set forth by Russel et al. [85], facilitated by SYCP3 co-staining where possible as outlined below. All quantification was carried out via direct visualization. Underlying data can be found in S1 Data.

## Antibody labeling

For colocalization studies, ADAD2, RNF17, and PIWIL1 were fluorescently labeled using Zenon Rabbit IgG Labeling Kits (Alexafluor 488 and Alexafluor 594, Thermofisher Scientific) as per manufacturer's instruction.

## Confocal visualization

Slides prepared for confocal visualization were processed as standard IFs up until mounting, with the same antigen retrievals and primary and secondary antibody concentrations (see above and S2 Table). Applications used 4 µM sections (or 8 µM as specified). After incubation with secondary antibody, slides were counterstained with DAPI (Sigma-Aldrich). DAPI dissolved in deionized water (20 µg/µL) was applied to tissue sections and incubated in a light-protected humid chamber for 20 minutes at room temperature. Light protected slides were then rinsed in running deionized water for 20 minutes and mounted with ProLong Glass Anti-fade mountant (Thermo Scientific) per manufacturer's instructions.

Slides were imaged on a Leica TCS SP8 tauSTED 3X with Lightning capabilities using the 100x objective (HC PL APO CS2 100x/1.40 OIL). Images taken of tubules at stages IX-XI [85]. Acquisition format was 1024x1024, speed 400hz, and a pinhole of 0.5. Line average was 4 for all channels. Z-stack step size was 0.13 µM. Z-stacks were captured with Lightning using Leica Application Suite X (LAS-X, Leica) and granule measurements were taken using the inbuilt quantification functions in LAS-X. Images and movies were color-combined and re-sliced using FIJI (Image J) [86].

## Granule size and colocalization quantification

Immunofluorescent images for granule analyses were processed and analyzed with FIJI (Image J). Single channel images at 400x magnification were captured on a custom-built Zeiss microscope, as described above. Each channel underwent processing before analyses. First, debris, tubules that were not the cross-section of interest, and any non-spermatocyte cells, were removed. Background subtraction was performed using the inbuilt tool. Granule sizes and counts were conducted using FIJI's inbuilt particle analysis tool. Briefly, each image was manually thresholded to select signal from background, then the number of granules of a specific size (5–20 pixels for small, 20–30 for intermediate, and 30 or greater for large granules) and circularity (0.8–1.0 for all) were counted and measured. Analyses were carried out at selected developmental stages (n = 3 tubules per stage, all mature spermatocytes in each cross section). Colocalization was assessed using JACoP (Just Another Colocalization Plugin) [87] for FIJI. Each channel threshold was manually set in JACoP. ADAD2 granules were compared by size to total PIWIL1 or COX IV signal, using the Manders' Colocalization Coefficient. Underlying data can be found in S1 Data.

### Tubule staging criteria

Stages of seminiferous tubule sections were determined according to the definitions outlined previously [85], along with a combination of SYCP3 and DAPI staining [88]. As both $Adad2^{M/M}$ and $Rnf17^{M/M}$ males do not complete spermatogenesis, staging was reliant on cell types present prior to ADAD2 expression, primarily preleptotene, leptotene, zygotene, and early pachytene spermatocytes.

## Supporting information

**S1 Fig. Mutation of *Adad2* or *Rnf17* leads to minimal abundances changes of the other.**
Western blot of ADAD2 or RNF17 in **A.** 42 dpp wildtype, $Adad2^{M/M}$, and $Rnf17^{M/M}$ whole testis protein (n = 3) demonstrating complete ADAD2 or RNF17 ablation in the respective genetic model and **B.** 21 dpp wildtype, $Adad2^{M/M}$, and $Rnf17^{M/M}$ whole testis protein (n = 3). Asterisks—RNF17 protein isoforms. Approximate molecular weight reported for each band. For loading controls, see S10 Fig.
(PDF)

**S2 Fig. ADAD2 and RNF17 share a similar developmental profile. A.** Western blot of ADAD2 and RNF17 in wildtype whole testis protein across neonatal and juvenile developmental time points demonstrating the similar developmental profile of ADAD2 and RNF17L. Asterisks—RNF17 protein isoforms. Approximate molecular weight reported for each band. For loading controls, see S10 Fig. **B.** Immunofluorescence of ADAD2 or RNF17 in wildtype testis across juvenile development. Images represent most mature seminiferous tubule sections at each age and demonstrate large ADAD2 and RNF17 granules forms by 15 dpp. Asterisks–small ADAD2 or RNF17 granules. Arrowheads–large ADAD2 or RNF17 granules. 200x magnification.
(PDF)

**S3 Fig. ADAD2's granular localization in pachytene spermatocytes requires RNF17. A.**
Immunofluorescence of ADAD2 across pachytene spermatocyte development in adult wildtype testes demonstrating small and large granule formation in mid-stage pachytene spermatocytes. **B.** ADAD2 immunofluorescence in $Adad2^{M/M}$ and $Rnf17^{M/M}$ developing pachytene spermatocytes. Note the non-specific ADAD2 signal observed in *Adad2* mutant round spermatids. Non-specific spermatid staining marked with an open arrowhead. Roman numerals–testis tubule cross-section stage (V containing early-stage pachytene spermatocytes, VII and VIII containing mid-stage pachytene spermatocytes, IX and X containing late-stage pachytene spermatocytes). Asterisks—small granules and arrowheads—large granules. Red—SYCP3, green—ADAD2, and blue—DAPI. 400x magnification.
(PDF)

**S4 Fig. Large and small ADAD2-RNF17 granules are molecularly distinct from one another and large ADAD2-RNF17 granules are unique from other defined granules. A.**
Co-immunofluorescence of ADAD2 and RNF17 using fluorophore labeled anti-ADAD2 and anti-RNF17 in $Adad2^{M/M}$ and $Rnf17^{M/M}$ adult testes demonstrating weak, cytoplasmically diffuse non-specific signal. Red–RNF17, green–ADAD2, and blue–DAPI. **B.** Immunofluorescence of DDX4 and ADAD2 or RNF17 in adult wildtype testes demonstrating neither ADAD2 nor RNF17 colocalize with DDX4. Red—DDX4, green—ADAD2 or RNF17, and blue—DAPI. **C.** Immunofluorescence against DDX25 and ADAD2 or RNF17 demonstrates that DDX25 colocalizes with some small ADAD2-RNF17 granules but not the large. Red—DDX25, green —RNF17, and blue—DAPI. Asterisks—small granules and arrowheads—large granules. All

images 630x magnification.
(PDF)

**S5 Fig. ADAD2 and RNF17 do not colocalize with PIWIL2 and their loss does not impact PIWIL protein localization. A.** Co-immunofluorescence of PIWIL2 with ADAD2 or RNF17 in adult wildtype testes demonstrating a lack of colocalization. Red—PIWIL2, green—ADAD2 or RNF17, and blue–DAPI. Asterisks—small ADAD2 or RNF17 granules. Arrowheads—large ADAD2 or RNF17 granules. 630x magnification. **B.** Immunofluorescence of PIWIL1 and PIWIL2 in *Adad2*$^{M/M}$ and *Rnf17*$^{M/M}$ adult testes by stage (as measured by SYCP3) showing no impact on PIWIL protein localization. Roman numerals–testis tubule cross-section stage. Red–SYCP3, green–PIWIL1 or PIWIL2, and blue–DAPI. 400x magnification.
(PDF)

**S6 Fig. ADAD2-RNF17 granule size and protein association changes across spermatocyte development. A.** ADAD2 granules per cell as a function of tubule cross section stage demonstrating a loss of small granules concurrent with an increase in large. Error bars–standard deviation. Asterisks (black–small granule comparisons, magenta–large granule comparisons)–significant by one-tailed t-test as compared to stage VII, *—p-value < 0.05, **—p-value < 0.001, ***—p-value < 0.0001. Mander's colocalization coefficients compared by stage and granule size **B.** ADAD2 localization with PIWL1 (asterisks–significant by two-tailed t-test as compared to stage VII, *—p-value < 0.05, **—p-value < 0.01, ***—p-value < 0.001) and **C.** ADAD2 localization with COX IV (significance calculated by two-tailed t-test. Asterisk—as compared to stage VII, *—p-value < 0.05. Pound sign–as compared to stage VIII, # < 0.05 and ## < 0.01).
(PDF)

**S7 Fig. ADAD2-RNF17 granules associate with the endoplasmic reticulum. A.** Co-immunofluorescence in adult wildtype testes of the nuclear membrane marker Lamin A/C and ADAD2 or RNF17 demonstrating neither ADAD2 nor RNF17 large granules colocalize with the nuclear membrane Red–Lamin A/C, green—ADAD2 or RNF17, and blue—DAPI. 630x magnification for above images. Co-immunofluorescence in adult wildtype testes of the endoplasmic reticulum markers. **B.** SERCA1 and **C.** PDI with ADAD2 or RNF17 showing clustering of the ER at large ADAD2 or RNF17 granules. Red–SERCA1 or PDI, green—ADAD2 or RNF17, and blue—DAPI. 400x magnification for B and C. Asterisks—small granules and arrowheads—large granules.
(PDF)

**S8 Fig. ADAD2:RNF17 double mutants display post-meiotic germ cell loss. A.** Western blot of 21 dpp wildtype and *Adad2*$^{M/M}$:*Rnf17*$^{M/M}$ whole testis lysate (n = 3) probed for ADAD2 and RNF17 confirming ablation of both proteins. Approximate molecular weight reported for each band. **B.** Adult wildtype and *Adad2*$^{M/M}$: *Rnf17*$^{M/M}$ testis tubule cross-sections stained with H&E demonstrating the range of tubule defects in double mutant testes, including significant post-meiotic germ cell loss. **C.** Average number of round spermatids per tubule per developmental stage in adult testes from wildtype and *Adad2*$^{M/M}$: *Rnf17*$^{M/M}$ animals (n = 3). Data are mean ± s.d. Significance was calculated using an unpaired, one-tailed Student's t-test (*$P < 0.05$, **$P < 0.005$, ***$P < 0.0005$).
(PDF)

**S9 Fig. Gel and Western blot confirmation of IP efficiency in representative wildtype and *Adad2*$^{M/M}$ immunoprecipitation (IP) samples. A.** SYPRO-Ruby stained SDS-PAGE gel. **B.** Western blot against ADAD2. Approximate molecular weight reported for each band.
(PDF)

**S10 Fig. Western blot loading controls.** SYPRO-Ruby stained membranes for blots shown in **A.** S1 Fig, **B.** S2 Fig, and **C.** S8 Fig showing equal loading across lanes.
(PDF)

**S1 Table. IP-MS identified peptides by replicate.**
(XLSX)

**S2 Table. Immunofluorescence Antibodies and conditions.** Antibodies used in this manuscript for immunofluorescence, their species, supplier, and product number, and the dilution used.
(XLSX)

**S1 Movie. The ADAD2 granule through multiple vertical planes.** ADAD2 granules Red—SYCP3, green—ADAD2. Slices 0.13 μM apart. 1000x magnification.
(MOV)

**S2 Movie. ADAD2 and RNF17 colocalization across multiple vertical planes.** Red—RNF17, green—ADAD2. Slices 0.13 μM apart. 1000x magnification.
(MOV)

**S1 Data. Underlying data.** For Fig 1A, 1B, 2A, 2C, and 9A and S6A–S6C Fig and S8C Fig.
(XLSX)

## Acknowledgments

The authors would like to thank current and previous members of the Snyder laboratory including Yeva Shamailova, Aditi Badrinath, Kelly Seltzer, Christopher Eddy, Gabriella Acoury, Gabrielle Vittor, and Megan Forrest for their support with animal husbandry, molecular analyses, and critical evaluation throughout the project. We would additionally like to thank Drs. Jeremy Wang and Xin Li for the kind gift of *Rnf17* mutant mice, the Human Genetics Institute of New Jersey Imaging Core for their support in confocal imaging, and Dr. Jessica Shivas and Daniel Jung for their assistance with confocal data acquisition and analyses.

## Author Contributions

**Conceptualization:** Lauren G. Chukrallah, Elizabeth M. Snyder.

**Data curation:** Lauren G. Chukrallah, Sarah Potgieter, Elizabeth M. Snyder.

**Formal analysis:** Lauren G. Chukrallah, Sarah Potgieter, Elizabeth M. Snyder.

**Funding acquisition:** Elizabeth M. Snyder.

**Investigation:** Lauren G. Chukrallah, Sarah Potgieter, Lisa Chueh, Elizabeth M. Snyder.

**Methodology:** Lauren G. Chukrallah, Sarah Potgieter, Elizabeth M. Snyder.

**Project administration:** Elizabeth M. Snyder.

**Supervision:** Elizabeth M. Snyder.

**Validation:** Lauren G. Chukrallah, Sarah Potgieter, Lisa Chueh.

**Visualization:** Sarah Potgieter.

**Writing – original draft:** Lauren G. Chukrallah, Elizabeth M. Snyder.

**Writing – review & editing:** Sarah Potgieter, Elizabeth M. Snyder.

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
