## [Decision Letter · Decision Letter 0]

22 Jan 2023

Dear Dr Snyder,

Thank you very much for submitting your Research Article entitled 'Two RNA binding proteins, ADAD2 and RNF17, interact to form novel meiotic germ cell granules required for male fertility' to PLOS Genetics.

The manuscript was fully evaluated at the editorial level and by independent peer reviewers. The reviewers appreciated the attention to an important problem, but raised some substantial concerns about the current manuscript. Based on the reviews, we will not be able to accept this version of the manuscript, but we would be willing to review a much-revised version. We cannot, of course, promise publication at that time.

If you decide to revise the manuscript for further consideration at PLOS Genetics, please aim to resubmit within the next 60 days, unless it will take extra time to address the concerns of the reviewers, in which case we would appreciate an expected resubmission date by email to plosgenetics@plos.org.

We are sorry that we cannot be more positive about your manuscript at this stage. Please do not hesitate to contact us if you have any concerns or questions.

Yours sincerely,

Wei Yan, MD, PhD

Guest Editor

PLOS Genetics

Gregory Barsh

Editor-in-Chief

PLOS Genetics

Your manuscript was reviewed by two expert reviewers. I encourage you to address the major concerns raised by the two reviewers and submit it back for further evaluation.

Reviewer's Responses to Questions

**Comments to the Authors:**

Reviewer #1: The study by Chukrallah et al. investigates the molecular composition and function of the cytoplasmic ribonucleoprotein granules, or germ granules, in meiotic cells. Electron microcopy and immunostaining analyses have revealed distinct type of granules in spermatocytes, and many of these granule types remain poorly characterized. It has been shown before that the Adenosine deaminase domain-containing protein 2, ADAD2, specifically localizes to the cytoplasmic granules meiotic germ cells, and that ADAD2 is required for haploid male germ cell differentiation. Here the authors show that ADAD2-granules are also positive for RNF17, a Tudor domain-containing protein implicated in the PIWI-interacting RNA (piRNA) pathway. ADAD2 and RNF17 also interacted in the testis, and the mutant mice for Adad2 and Rnf17 were demonstrated to share very similar spermatogenic phenotype. The interaction was important for the germ granule localization, and ADAD2 failed to localize to the granules in the absence of RNF17 and vice versa. The authors also showed that there is heterogeneity in ADAD2-RNF17 positive granules with some of them co-localizing with PIWI proteins, while others co-localizing with a translational regulator NANOS1 to cup-shaped structures associated with the endoplasmic reticulum.

Although the study provides interesting information about the different germ granules in spermatocytes, it remains descriptive, mostly concentrating on characterizing the protein localizations without novel mechanistic data on the germ granules or ADAD2 functions. Both mutant mouse lines have already been published in earlier studies, and the granular cytoplasmic localization of ADAD2 and RNF17 have also been published, including the observation that ADAD2 granules do not overlap with DDX25. It is an interesting result that ADAD2 and the Tudor protein RNF17 co-localize to the granules distinct from the IMC or CB, and appear to regulate the similar phase of spermatogenesis. However, without further functional characterization the study remains quite thin, and it remains unclear, for example, if ADAD2 is involved in the piRNA pathway, e.g. in the suppression of the ping-pong amplification, a process that has been shown to be regulated by RNF17.

Specific comments:

- The title states that the ADAD2/RNF17 germ granules are required for normal male fertility. These proteins do co-localize to the same cytoplasmic granules, but they also localize elsewhere – therefore, it feels premature to say that these granules are required for male fertility, and I think it can only be concluded that ADAD2 and RNF17 are required for male fertility.

- Abstract: the exact point of the following sentence is difficult to understand: “Lastly, a double Adad2-Rnf17 mutant model demonstrated loss of ADAD2-RNF17 granules themselves, as opposed to loss of either ADAD2 or RNF17, is the likely driver of the Adad2 and Rnf17 mutant phenotypes.” Should be clarified. Can you really make a conclusion that the loss of ADAD2-RNF17 granules causes the phenotype, is it also possible that the loss of ADAD2-RNF17 interplay in general, inside or outside the granules, causes the phenotype? Have you studied the granules containing ADAD2 and RNF17 by electron microscopy in the mutant germ cells, do you know if the whole granule structure is disrupted, or if the granules are still there even in the absence of these proteins?

- line 55: spell out “piRNA” when it appears for the first time

- line 55-56: piRNAs have also important role in transposon silencing, this function should be mentioned.

- line 57: maybe would be better to say “the primary piRNA-binding proteins” rather than “biogenesis factors”. Many other important factors are involved in the piRNA biogenesis, and PIWI protein also mediate piRNA functions after the biogenesis.

- line 72: spell out ADAD2

- Western blot images: The molecular weight marker sizes have to be shown in western blots. Please also explain what are the samples (Input Flow-through, Wash, IP) in the figure legend of Fig. 1C.

- The mass spec results should be provided as a supplementary file, including the information about the peptide hits in each sample.

- Fig. 3A is missing the information on which antibodies were used in the immunostaining.

- Fig. 4B,C: Co-localization of ADAD2 and RNF17 with PIWIL1 and PIWIL2 is somewhat difficult to interpret. To me it looks like ADAD2/RNF17 and PIWIL2 granules are distinct, and ADAD2/RNF17 would not co-localize with PIWIL2 to same granules, but ADAD2/RNF17 and PIWIL2 granules are just found often very close to each other?

- Fig. 6A: what is the strong ADAD2-positive structure in round spermatids? Does the antibody give (unspecific?) acrosomal staining?

Reviewer #2: In this manuscript, the authors identified RNF17 as a major ADAD2 binding partner and together forming a novel type of germ granule in meiotic male germ cells. Through a meticulous study of ADAD2 and RNF17 positive germ cell granules by imaging and mouse genetic models, they show the dynamic formation of subtypes of granules and potential functional involvement in germ cell development. Germ granule dynamics are understudied in meiotic cells. This study paves the way for understanding new connections of germ granules to different cellular pathways critical for germ cell differentiation.

Major comments:

The round spermatid chromocenter defect in Adad2 and RNF17 KO mice is intriguing. How would the authors explain RBPs like ADAD2 and RNF17 affect heterochromatin formation in the nucleus. Is there a genetic link between ADAD2/RNF17 and chromocenter regulators such as TLF and BRDT [ref 27, 28]?

Line 215: The authors stated that previous analysis of ADAD2 granules has demonstrated ADAD2 does not colocalize with DDX25 [23]. But in fact, reference [23] indicates that ADAD2 granules showed partial overlap with DDX25 in late pachytene and diplotene spermatocytes (Fig. 5C of Elizabeth Snyder et al. 2020 [23]). The authors need to add an image of ADAD2 and DDX25 co-staining in late pachytene or diplotene (stage IX-XI). In Fig S4C, there are some granules with yellow-orange color (overlap of green and red) which means that RNF17 and DDX25 partially overlap.

Different types of granules are still vaguely defined. Granules are in a dynamic state of aggregation and dissociation. Some groups also defined granules in late pachytene and diplotene spermatocytes as chromatoid body precursors. Without immune-EM, it is still not convincing to conclude that large ADAD2/RNF17 granules are the so-called cluster of 30 nm particles.

What reason could the ADAD2-RNF17 granule be interacting with the ER? Dive further in this in the discussion would be helpful.

Minor comments:

Line 113, Line 125: in vitro should be in vivo.

Fig S1 and Fig S2A: These WB images need to have internal control such as β-ACTIN or GAPDH to ensure consistent sample loading.

Fig 3A: the authors should label RNF17, SYCP3 and DAPI on the image.

Typos:

Line 143: add comma after “similar profile”; Line 298: change “therefor” to “therefore”; Line 461: “Adad2:Rn17” should read “Adad2:Rnf17”.

**Have all data underlying the figures and results presented in the manuscript been provided?**

Reviewer #1: Yes

Reviewer #2: Yes

PLOS authors have the option to publish the peer review history of their article (what does this mean?). If published, this will include your full peer review and any attached files.

Reviewer #1: No

Reviewer #2: No

---

## [Decision Letter · Decision Letter 1]

17 Jun 2023

Dear Dr Snyder,

We are pleased to inform you that your manuscript entitled "Two RNA binding proteins, ADAD2 and RNF17, interact to form a heterogeneous population of novel meiotic germ cell granules with developmentally dependent organelle association" has been editorially accepted for publication in PLOS Genetics. Congratulations!

Yours sincerely,

Wei Yan, MD, PhD

Guest Editor

PLOS Genetics

Gregory Barsh

Editor-in-Chief

PLOS Genetics

Comments from the reviewers (if applicable):

Reviewer's Responses to Questions

**Comments to the Authors:**

Reviewer #1: The authors have made a big effort to respond to all my comments and concerns. A lot of new data was added and the manuscript was revised accordingly, including the revision of some conclusions. Although no mechanistic insight on ADAD2 and RNF17 function was revealed, this study provides very detailed and thorough characterization of the ADAD2/RNF17 granules, revealing important information about the composition and dynamics of the germ granules in meiotic cells.

Reviewer #2: The authors have made significant improvement and satisfactorily addressed all my questions.

This reviewer found the revised manuscript title appropriate. In light of two very recent publications on ADAD2 and RNF17 during revision, this manuscript is distinct and complementary which warrants a timely publication.

**Have all data underlying the figures and results presented in the manuscript been provided?**

Reviewer #1: Yes

Reviewer #2: Yes

PLOS authors have the option to publish the peer review history of their article (what does this mean?). If published, this will include your full peer review and any attached files.

Reviewer #1: No

Reviewer #2: No

**Data Deposition**

http://datadryad.org/submit?journalID=pgenetics&manu=PGENETICS-D-22-01297R1

**Press Queries**

---

## [Editor Report · Acceptance letter]

5 Jul 2023

PGENETICS-D-22-01297R1 

Two RNA binding proteins, ADAD2 and RNF17, interact to form a heterogeneous population of novel meiotic germ cell granules with developmentally dependent organelle association 

Dear Dr Snyder, 

We are pleased to inform you that your manuscript entitled "Two RNA binding proteins, ADAD2 and RNF17, interact to form a heterogeneous population of novel meiotic germ cell granules with developmentally dependent organelle association" has been formally accepted for publication in PLOS Genetics! Your manuscript is now with our production department and you will be notified of the publication date in due course.

With kind regards,

Zsofia Freund

PLOS Genetics

On behalf of:
